# Attention on the Sphere

**Boris Bonev**[†*], **Max Rietmann**[†*], **Andrea Paris, Alberto Carpentieri, Thorsten Kurth**[†*]
NVIDIA Corporation, 95051 Santa Clara, CA, USA
{bbonev, mrietmann, aparis, acarpentieri, tkurth}@nvidia.com

## Abstract

We introduce a generalized attention mechanism for spherical domains, enabling Transformer architectures to natively process data defined on the two-dimensional sphere - a critical need in fields such as atmospheric physics, cosmology, and robotics, where preserving spherical symmetries and topology is essential for physical accuracy. By integrating numerical quadrature weights into the attention mechanism, we obtain a geometrically faithful spherical attention that is approximately rotationally equivariant, providing strong inductive biases and leading to better performance than Cartesian approaches. To further enhance both scalability and model performance, we propose neighborhood attention on the sphere, which confines interactions to geodesic neighborhoods. This approach reduces computational complexity and introduces the additional inductive bias for locality, while retaining the symmetry properties of our method. We provide optimized CUDA kernels and memory-efficient implementations to ensure practical applicability. The method is validated on three diverse tasks: simulating shallow water equations on the rotating sphere, spherical image segmentation, and spherical depth estimation. Across all tasks, our spherical Transformers consistently outperform their planar counterparts, highlighting the advantage of geometric priors for learning on spherical domains.

## 1 Introduction

The two-dimensional sphere embedded in three dimensions ($S^2$) plays a crucial role in a variety of scientific and engineering domains such as geophysics, cosmology, chemistry, and computer graphics. All of these domains require the processing of functions defined on the surface of the sphere. With the advent and increased success of machine-learning in these domains, there is a growing demand to generalize state-of-the-art architectures to spherical data [13, 18, 19, 34, 6]. This is particularly important in scientific applications which require the solution of time-dependent partial differential equations which are often approximated with auto-regressive models. These models are particularly sensitive to spurious artifacts which arise from a non-geometrical treatment as these tend to build up and compromise stability [6].

While spherical convolutional neural networks (CNNs) and other geometric deep learning models have advanced the state of the art for learning on the sphere by respecting its symmetries and topology, these architectures are primarily designed for capturing local interactions and equivariant representations [13, 34, 30]. However, many scientific and engineering problems on the sphere, such as weather and climate modeling, 360°perception, and cortical surface analysis, require modeling complex, long-range dependencies that are not efficiently handled by convolutional approaches [7, 3, 9]. Transformers, with their global attention mechanism, have revolutionized learning in Euclidean domains by enabling flexible, data-driven modeling of both local and global relationships

---

[*]Corresponging author.
[†]Equal contribution.

[15, 38, 4, 24]. Extending Transformers to spherical domains promises to combine the geometric fidelity of spherical models with the expressive power of attention, enabling new capabilities for spherical data analysis.

Recent works have begun to explore this direction, demonstrating that Transformer-based models tailored to spherical geometry can outperform traditional methods on tasks such as depth estimation or semantic segmentation by directly leveraging spherical geometry in their design [3, 10]. However, due to their reliance on permutation equivariance on icosahedral grids, they fail to capture the full rotational equivariance associated with the sphere. Other works have successfully applied Euclidean Transformer architectures to weather forecasting on the sphere [4], however, they show deterioration in longer auto-regressive rollouts [26]. This motivates the development of spherical attention mechanisms that operate natively on the sphere, unlocking the full potential of Transformer architectures for scientific and engineering applications involving spherical data.

**Our contribution**   We generalize the attention mechanism to the spherical domain. To respect rotational symmetry, we derive a continuous formulation on the sphere for both global attention and neighborhood attention mechanisms. A discrete formulation is then derived using quadrature rules to approximate kernel integrals, thus ensuring approximate equivariance w.r.t. three-dimensional rotations. Both spherical attention variants are implemented in PyTorch, including custom CUDA extensions of the spherical neighborhood attention. We demonstrate the effectiveness of our approach by generalizing standard Transformer architectures to the spherical domain using our formulation and scoring them against their Euclidean counterparts in three different spherical benchmark problems.

Our implementation and code to reproduce experiments are available in the open-source library torch-harmonics.

## 2   Related work

**Transformers**   Attention mechanism and Transformer architectures were initially developed for sequence modeling applications such as language models [37] and later applied to vision tasks in the Euclidean domain [15]. This has proven to be an effective method despite the lack of inductive biases such as locality [4, 24]. The attention mechanism is equivariant with respect to permutations of the input tokens and therefore lacks any notion of the underlying topology of the domain. This makes position embeddings necessary, which introduce the notion of locality. A notable advantage of the attention mechanism is its ability to capture long-range interactions. However, this makes it prohibitively expensive for large resolutions due to its $\mathcal{O}(N^2)$ asymptotic complexity, especially for multi-dimensional data where the number of pixels $N$ grows exponentially with dimension limiting the resolution.

**Neighborhood Transformers**   Hassani et al. [22] address this by simultaneously introducing the notion and inductive bias of locality into the architecture. This is achieved by using the neighborhood attention mechanism. In this approach, the attention layer is constrained to a neighborhood of grid points around the query point. This has great computational benefits as it lowers the asymptotic complexity of the attention mechanism to $\mathcal{O}(kN)$ where $k$ is the kernel size of the neighborhood. While this method is highly efficient [23], it is generally only formulated for equidistant Cartesian grids. For spherical problems, this leads to distorted neighborhoods on spherical geometries.

**Equivariant architectures**   Several convolutional architectures address this problem and capture the geometric properties alongside $SO(3)$ rotational symmetry [12, 17, 13, 11, 34, 6]. it has not been addressed by Transformer architectures yet. Finally, Assaad et al. [2] introduce a rotationally equivariant Transformer architecture in the context of vector neurons using the Frobenius norm.

**Graph equivariant Transformers**   Various equivariant transformer architectures have been proposed in the literature. Notable examples are the $SE(3)$-transformer [20] and the Equiformer architecture [29], and related architectures [39, 8]. However, these architectures focus on vector-valued features as inputs, rather than entire functions defines over a subdomain. Examples are positions of atoms in molecules and LiDAR point clouds. As such, the attention mechanisms in these architectures aim to preserve both distances of positions with each other and the relative orientation of the vectors.

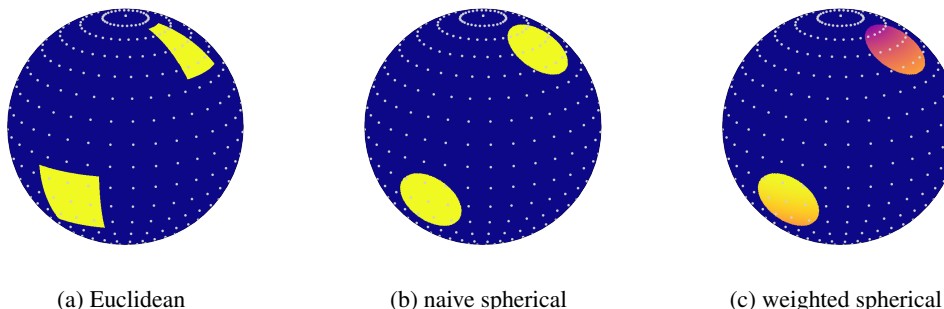

| (a) Euclidean | (b) naive spherical | (c) weighted spherical |

Figure 1: Neighborhood attention on the sphere: (a) Euclidean neighborhoods use planar distances, distorting receptive fields near poles. (b) Naive spherical attention uses geodesic neighborhoods but equally weights points, oversampling points and breaking SO(3) symmetry. (c) Our method introduces quadrature weights to account for non-uniform sampling density, preserving rotational equivariance regardless of grid alignment through proper discretization of the continuous formulation.

**Graph transformers applied to spherical functions**  Several graph-transformer models have been introduced that operate on icosahedral discretizations of the sphere [10, 9, 3], enabling the application of transformer architectures to function-valued data on spherical domains. While this approach has shown strong performance for spherical signals, it relies on the standard attention mechanism, which ensures only permutation equivariance with respect to the underlying graph. Consequently, these architectures exploit discrete graph symmetries rather than the continuous rotational symmetry of the sphere itself.

## 3  Method

### 3.1  Attention mechanism and kernel regression

We revisit the motivation behind the original attention mechanism. For two one-dimensional signals $q, k : \mathbb{R}^+ \to \mathbb{R}$, we can compute the correlation function $C[q, k](x, x') = \text{corr}(q(x), k(x'))$ using the Pearson correlation $\text{corr}(\cdot, \cdot)$, which measures the statistical dependence between signal pairs as a function of the point-pair $(x, x')$. Many stochastic processes, images and other spatial signals have distinct auto-correlation functions, which can be modeled using a kernel function $\alpha(x, x')$.

This analogy between correlation functions and kernel functions is exploited in nonparametric regression techniques such as kernel regression. For query/key signals $q, k : \mathbb{R}^+ \to \mathbb{R}^d$, and value signal $v : \mathbb{R}^+ \to \mathbb{R}^e$, the kernel regressor is given by

$$r(q(x)) = \int_{\mathbb{R}^+} \frac{\alpha(q(x), k(x'))}{\int_{\mathbb{R}^+} \alpha(q(x), k(x')) \, \mathrm{d}x'} \, v(x') \, \mathrm{d}x', \tag{1}$$

where $\alpha(\cdot, \cdot)$ is a suitable kernel function. For instance, by choosing $q(x) = x$, $k(x') = x'$ and the Gaussian kernel $\alpha(x, x') = \exp\big(-(x - x')^2/h^2\big)$ of width $h$, we recover the radial-basis regressor.

The attention mechanism [37] generalizes this concept further through learnable kernel transformations. The continuous-domain attention mechanism

$$\text{Attn}[q, k, v](x) = \int_{\mathbb{R}^+} A[q, k](x, x') \, v(x') \, \mathrm{d}x', \tag{2a}$$

$$A[q, k](x, x') = \frac{\exp\big(q^T(x) \cdot k(x')/\sqrt{d}\big)}{\int_{\mathbb{R}^+} \exp\big(q^T(x) \cdot k(x')/\sqrt{d}\big) \, \mathrm{d}x'}, \tag{2b}$$

corresponds to exponential kernel smoothing, where the exponential kernel $\alpha(q, k) = \exp(-q^T k/\sqrt{d})$ measures similarity between query-key pairs while the denominator normalizes attention weights as probability distributions. The linearized variant [36] omits softmax normalization,

analogous to using unnormalized kernel weights in regression numerators. Modern attention mechanisms extend this kernel method by learning adaptive correlation patterns which are parameterized by the query and key weights, rather than relying on predefined kernel functions.

By discretizing the continuous attention mechanism (2) on the sequence $\{x_i\}_{i=1}^{N_{\text{seq}}}$, we recover the standard attention mechanism

$$\text{Attn}[q, k, v](x) = \sum_{j=1}^{N_{\text{seq}}} \frac{\exp\left(q^T(x_i) \cdot k(x_j)/\sqrt{d}\right)}{\sum_{l=1}^{N_{\text{seq}}} \exp\left(q^T(x_i) \cdot k(x_l)/\sqrt{d}\right)} \, v(x_j), \tag{3}$$

commonly found in sequence modeling [37] and other applications. This formulation is permutation-equivariant, in the sense that a permutation of the tokens $q(x_i)$, $k(x_i)$ and $v(x_i)$ will result in the same output but permuted.

## 3.2 Attention on the sphere

The generalization of the attention mechanisms for spherical data requires a proper treatment of spherical geometry and topology. To do so, we change the domain in the continuous formulation (2) to the sphere and obtain

$$\text{Attn}_{S^2}[q, k, v](x) = \int_{S^2} A_{S^2}[q, k](x, x') \, v(x') \, \mathrm{d}\mu(x'), \tag{4a}$$

$$A_{S^2}[q, k](x, x') = \frac{\exp\left(q^T(x) \cdot k(x')/\sqrt{d}\right)}{\int_{S^2} \exp\left(q^T(x) \cdot k(x'')/\sqrt{d}\right) \mathrm{d}\mu(x'')}, \tag{4b}$$

for inputs $q, k : S^2 \to \mathbb{R}^d$, $v : S^2 \to \mathbb{R}^e$. $\mu : S^2 \to \mathbb{R}_0^+$ denotes the invariant Haar measure on the sphere and ensures that the integrals in (4) remain unchanged under three-dimensional rotations $R \in \text{SO}(3)$ of the variables of integration. This makes the attention mechanism (4) equivariant, i.e., $\text{Attn}_{S^2}[q, k, v](R^{-1}x) = \text{Attn}_{S^2}[q', k', v'](x)$ for rotated signals $q'(x) = q(R^{-1}x)$, $k'(x) = k(R^{-1}x)$, $v'(x) = v(R^{-1}x)$.

A local, neighborhood attention variant of (4) can be derived by choosing a compactly supported kernel $\alpha$. We define

$$N_{S^2}[q, k](x, x') = \frac{\mathbb{1}_{D(x)}(x') \, \exp\left(q^T(x) \cdot k(x')/\sqrt{d}\right)}{\int_{S^2} \mathbb{1}_{D(x)}(x'') \, \exp\left(q^T(x) \cdot k(x'')/\sqrt{d}\right) \mathrm{d}\mu(x'')}, \tag{5}$$

where $\mathbb{1}_{D(x)}(x')$ is the indicator function of the spherical disk $D(x) = \{x' \in S^2 | d(x, x')_{S^2} \le \theta_{\text{cutoff}}\}$ centered at $x$. This disk is determined by the geodesic distance (great circle/Haversine distance) $d(\cdot, \cdot)_{S^2}$ and the hyperparameter $\theta_{\text{cutoff}}$ which controls the size of the neighborhood. By replacing the attention operator $A_{S^2}$ with $N_{S^2}$, we obtain the spherical neighborhood attention

$$\text{NAttn}_{S^2}[q, k, v](x) = \int_{S^2} N_{S^2}[q, k](x, x') \, v(x') \, \mathrm{d}\mu(x'), \tag{6}$$

which retains the equivariance property of (4).

To obtain discrete attention mechanisms, we discretize the spherical domain with a set of grid points $\{x_i \in S^2 \text{ for } i = 1, 2, \ldots, N_{\text{grid}}\}$ and apply a suitable quadrature formula $\omega_i := \omega(x_i)$, such that

$$\int_{S^2} u(x) \, \mathrm{d}x \approx \sum_{i=1}^{N_{\text{grid}}} u(x_i) \, \omega_i. \tag{7}$$

This allows a numerical evaluation of the integrands in (4) and (6) giving us the discrete full and neighborhood attention (see Appendix B.3 for additional detail).

Figure 1 visualizes our approach to neighborhood attention on spherical data. By deriving the continuous formulation through quadrature rules and rigorously incorporating geodesic distances, we establish our architecture as a neural operator. This formulation enables consistent evaluation

across arbitrary discretizations while preserving geometric structure. Crucially, the quadrature-based implementation ensures approximate SO(3) equivariance, maintaining transformation consistency under 3D rotations. To validate the equivariance of our implementation, we conduct a convergence test in Appendix D that demonstrates convergence of the approximate equivariance error in relation to the interpolation error as resolution is increased.

## 3.3 Implementation

We implement both spherical attention mechanisms for equiangular and Gaussian grids and make them publicly available in torch-harmonics, a library for machine learning and signal processing on the sphere. The global spherical attention may be efficiently implemented using the PyTorch scaled dot product attention implementation by leveraging the attention mask to incorporate the quadrature weights. The neighborhood variant, on the other hand, requires a custom CUDA implementation due to the non-uniform memory access patterns implied by the sparse attention support. Details regarding the implementation and expressions of input gradients are provided in Appendix B.4.

## 3.4 Spherical Transformer architecture

We apply the spherical attention mechanisms derived in Section 3 in the context of popular Transformer architectures. To this end, we modify the Vision Transformer (ViT) [15] to obtain a fully spherical architecture that attains equivariance properties. To do so, we replace non-spherical operations with spherical counterparts and derive the $S^2$ Transformer and $S^2$ neighborhood Transformer architectures (see Figure 2). A similar approach can be taken to obtain spherical variants of the SegFormer architecture [38]; i.e. the $S^2$ SegFormer and $S^2$ neighborhood SegFormers, respectively. Due to their continuous formulations, all of these architectures are neural operators [28], allowing them to be realized on arbitrary discretizations of the sphere, that permit a suitable quadrature rule.

In the following, we outline the methodology for deriving the spherical Transformers.

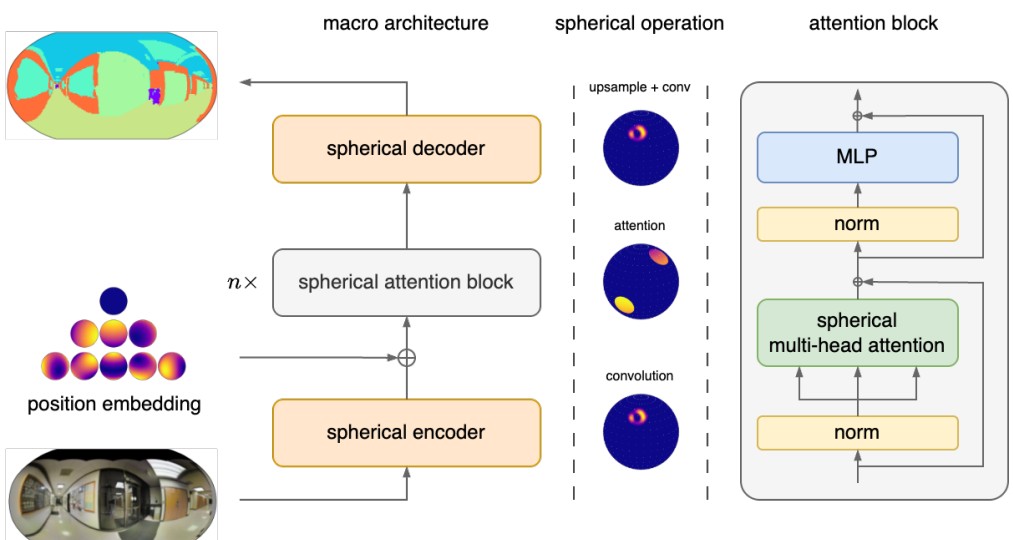

Figure 2: Illustration of the spherical Transformer architecture.

**Encoding**  Vision Transformers typically use patch embeddings which can be regarded as convolutions with a special choice of pre-set, sparse weights. As such, we employ a convolutional embedding scheme based on the discrete-continuous spherical convolutions [34]. Unlike standard patch embeddings, our approach leverages spherical convolutions to ensure that the resulting representation remains a function defined on $S^2$ preserving equivariance properties of the model with respect to rotations. Moreover, the discrete-continuous formulation makes these convolutions neural operators [30], enabling flexible discretization.

**Spherical attention blocks** The latent representation is refined by a sequence of $n$ spherical attention blocks (Fig. 2, right). Each block applies spherical multi-head attention followed by a multilayer perceptron (MLP), both with residual connections and preceded by instance normalization, forming a pre-norm architecture. Positional embeddings (defined in Section 3.4) are added before the multi-head attention step. The spherical attention can operate either globally across the entire sphere, as defined in Eq. (4), or locally, as shown in Eq. (6).

**Decoding** The final latent is passed through a decoder, which mirrors the encoding step. Bilinear spherical interpolation is performed to upsample the signal to the full resolution. This is followed by a spherical discrete-continuous convolution, which produces the output.

**Position embeddings** Transformer architectures typically incorporate positional embeddings to provide information about the location of each input token. This is necessary because the attention mechanism (3) is inherently permutation-invariant and thus unaware of spatial structure. We experimented with several possibilities including learnable [15], sequence-based [37], and spectral position embeddings [35]. In the end, we chose the spectral embedding, where the $k$-th channel is determined by the real-valued spherical harmonics $Y_\ell^m(x)$, with $\ell = \lfloor \sqrt{k} \rfloor$ and $m = k - \ell(\ell + 1)$.

The resulting spherical Transformer architecture is naturally a neural operator, as all spatial operations can be formulated in the continuous domain, as well as for arbitrary grid-resolutions which permit the accurate numerical computation of integrals over the sphere. Finally, the same design modifications are applied to the SegFormer architecture [38] to obtain the spherical SegFormer and spherical neighborhood SegFormer. These architectures are made available as part of torch-harmonics.

# 4 Experiments

We evaluate our method by adapting the popular Vision Transformer (ViT) and SegFormer architecture to equirectangular grids on the spherical domain. We follow the procedure described in Section 3.4, replacing the attention and convolution layers with their spherical counterparts while maintaining equivalent architectural configurations, such as depth and embedding size. In particular, receptive fields of neighborhood attention mechanisms and convolutions are carefully chosen to match the covered areas. This allows for a fair comparison of models, as models retain similar parameter counts, as listed in Table 5 in the appendix. A comprehensive study comparing spherical Transformers to their corresponding Euclidean baseline is carried out for both global and local (neighborhood) attention variants. Implementation details are provided in Appendix C.1.

## 4.1 Segmentation on the sphere

The Stanford 2D3DS Dataset [1] provides spherical images taken indoors in a university setting. It provides a total of 1621 RGB images along with a number of output pairs including segmentation data containing 14 classes, such as *clutter*, *ceiling*, *window*, etc. For the purpose of training, we downsample the data to a resolution of 128x256 and split the dataset into 95% train, 2.5% test and 2.5% validation parts. Training is carried using uniform weighted cross entropy loss for 200 epochs using the ADAMW optimizer [31], a learning rate of $0.5 \times 10^{-4}$ and a cosine scheduler. Furthermore, to reduce overfitting, weight decay and dropout path rate are both set to $0.1$. On a single NVIDIA RTX 6000 Ada, training took between 12 and 215 minutes depending on the respective models. The results are validated using the Accuracy (Acc) and Intersection over Union (IoU) metrics [25] on the validation dataset which contains 35 samples. For a detailed discussion refer to Appendix A.2.

Table 1 reports the results of our experiments. For both transformer and SegFormer models, we see the spherical variants outperforming their Euclidean counterparts, where the Transformer models have compatible hyperparameters across spherical and euclidean variants. We train EGFormer [40] and SphericalCNN [14] to serve as additional geometric baselines. We note that the EGFormer seems to overfit and may require different hyperparameters than the other architectures. Moreover, we report results for the SphereUFormer [3] alongside high-resolution variants of our neighborhood SegFormers with double the embedding dimension. We observe that our $S^2$ SegFormer with a locally supported spherical attention mechanism outperforms the other models in the validation metrics.

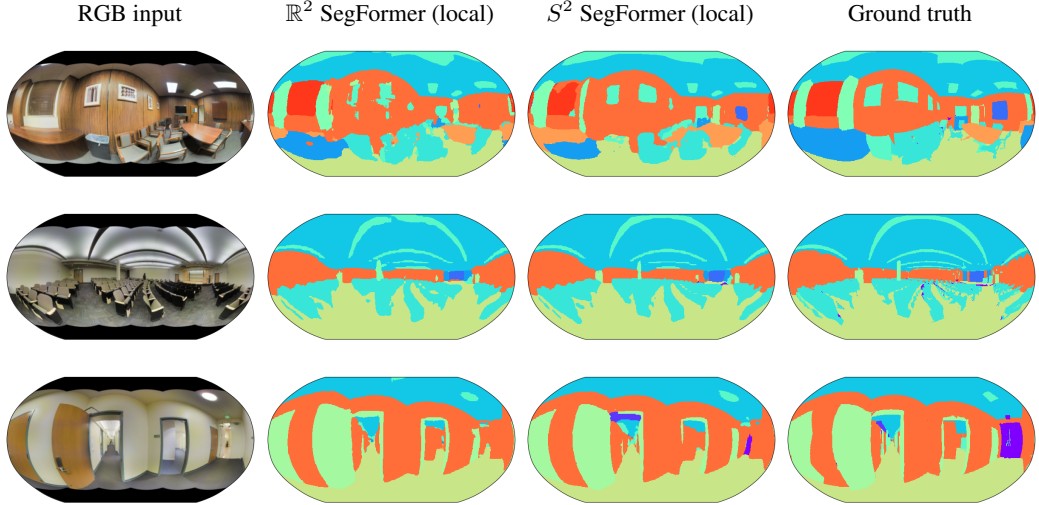

|  |  |  |  |
| RGB input | $\mathbb{R}^2$ SegFormer (local) | $S^2$ SegFormer (local) | Ground truth |

Figure 3: Samples from the segmentation dataset with Euclidean and $S^2$ neighborhood SegFormers.

Table 1: Numerical results on the segmentation dataset. Cross entropy training and validation loss alongside Intersection-over-Union (IoU) and Accuracy (Acc) metrics are reported.

| Model | Training loss ↓ | Validation loss ↓ | IoU ↑ | Acc ↑ |
|---|---|---|---|---|
| | Results on 128×256 data | | | |
| EGFormer [40] | **0.203** | $1.29 \pm 0.487$ | $0.586 \pm 0.106$ | $0.962 \pm 0.012$ |
| SphericalCNN [14] | 0.382 | $0.775 \pm 0.300$ | $0.622 \pm 0.099$ | $0.966 \pm 0.011$ |
| $\mathbb{R}^2$ Transformer | 0.325 | $1.041 \pm 0.812$ | $0.593 \pm 0.151$ | $0.962 \pm 0.018$ |
| $S^2$ Transformer | 0.312 | $1.016 \pm 0.499$ | $0.588 \pm 0.111$ | $0.962 \pm 0.013$ |
| $\mathbb{R}^2$ Transformer (local) | 0.344 | $0.854 \pm 0.352$ | $0.619 \pm 0.103$ | $0.966 \pm 0.011$ |
| $S^2$ Transformer (local) | 0.298 | $0.852 \pm 0.369$ | $0.622 \pm 0.106$ | $0.966 \pm 0.012$ |
| $\mathbb{R}^2$ Segformer | 0.296 | $0.796 \pm 0.363$ | $0.636 \pm 0.109$ | $0.967 \pm 0.012$ |
| $S^2$ SegFormer | 0.307 | $0.693 \pm 0.303$ | $0.657 \pm 0.106$ | $0.970 \pm 0.011$ |
| $\mathbb{R}^2$ Segformer (local) | 0.288 | $0.782 \pm 0.334$ | $0.637 \pm 0.101$ | $0.968 \pm 0.011$ |
| $S^2$ SegFormer (local) | 0.320 | $\mathbf{0.667 \pm 0.309}$ | $\mathbf{0.667 \pm 0.110}$ | $\mathbf{0.971 \pm 0.012}$ |
| | Results on 256×512 data | | | |
| SphereUFormer [3] | - | - | 0.722 | 0.886 |
| $\mathbb{R}^2$ Segformer (local, large) | **0.184** | $0.740 \pm 0.858$ | $0.712 \pm 0.153$ | $0.974 \pm 0.017$ |
| $S^2$ SegFormer (local, large) | 0.189 | $\mathbf{0.554 \pm 0.550}$ | $\mathbf{0.748 \pm 0.126}$ | $\mathbf{0.979 \pm 0.013}$ |

Figure 3 depicts results from the best performing local $S^2$ SegFormer architecture alongside its Euclidean counterpart. It demonstrates an improvement in the quality of segmentation masks, when compared to the ground truth target.

## 4.2 Depth estimation on the sphere

The Stanford 2D3DS dataset provides depth maps alongside segmentation data, which we use to train our models on the depth estimation task. Models are trained for 100 epochs using the $\mathcal{L}_{\text{depth}}$ objective functions (16) which combines $L_1$ and Sobolev $W^{1,1}$ losses, to better preserve image sharpness. The top and bottom 15% are masked out as there is no valid target data in this area. Training times vary between 11 and 220 minutes on a single NVIDIA RTX6000 Ada GPU depending on the architecture.

The numerical results alongside individual samples from the Transformer architectures are provided in Table 2 and Figure 4. We observe improved validation losses and $W^{1,1}$ error for spherical architectures (see Appendix A.2 for detail on the losses and norms).

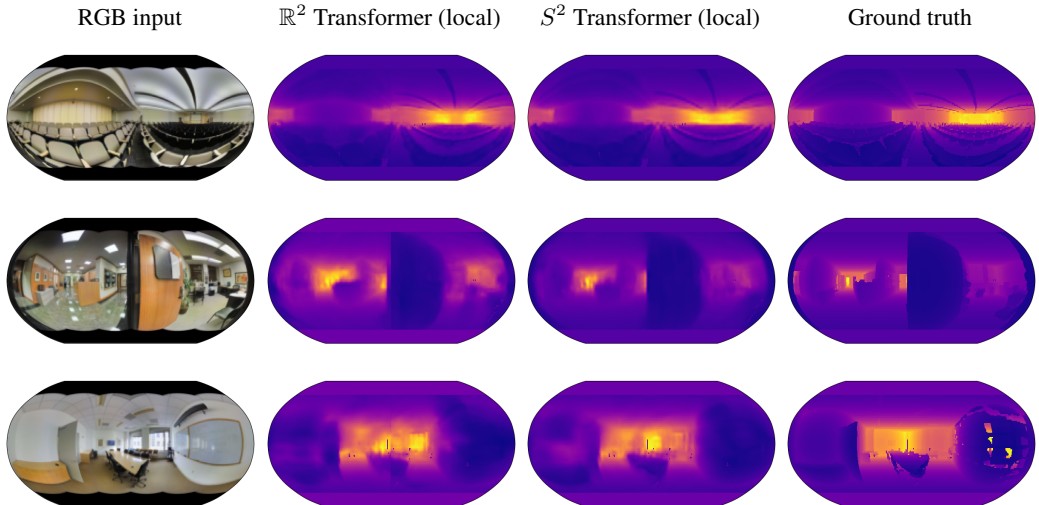

Figure 4: Sample predictions from our transformers alongside ground truth data from the spherical depth estimation task.

Table 2: Numerical results on the depth estimation dataset. Training and validation errors, alongside errors for various cartesian Transformer variants and their spherical counterparts.

| Model | Training loss ↓ | Validation loss ↓ | $L_1$ error ↓ | $W^{1,1}$ error ↓ |
|---|---|---|---|---|
| $\mathbb{R}^2$ Transformer | 0.954 | $0.982 \pm 0.386$ | $0.215 \pm 0.084$ | $7.67 \pm 3.22$ |
| $S^2$ Transformer | 0.978 | $0.993 \pm 0.381$ | $0.234 \pm 0.086$ | $\mathbf{7.59 \pm 3.23}$ |
| $\mathbb{R}^2$ Transformer (local) | 1.014 | $1.023 \pm 0.370$ | $0.237 \pm 0.079$ | $7.86 \pm 3.17$ |
| $S^2$ Transformer (local) | 0.978 | $0.976 \pm 0.354$ | $0.209 \pm 0.068$ | $7.68 \pm 3.12$ |
| $\mathbb{R}^2$ SegFormer | 1.037 | $0.980 \pm 0.371$ | $0.169 \pm 0.058$ | $8.11 \pm 3.22$ |
| $S^2$ SegFormer | 1.005 | $\mathbf{0.935 \pm 0.369}$ | $\mathbf{0.165 \pm 0.057}$ | $7.70 \pm 3.23$ |
| $\mathbb{R}^2$ SegFormer (local) | 1.055 | $1.002 \pm 0.374$ | $0.174 \pm 0.061$ | $8.27 \pm 3.25$ |
| $S^2$ SegFormer (local) | 1.014 | $0.948 \pm 0.368$ | $0.174 \pm 0.061$ | $7.75 \pm 3.21$ |
| additional baselines | | | | |
| Spherical CNN [14] | 1.019 | $1.098 \pm 0.412$ | $0.315 \pm 0.123$ | $7.83 \pm 3.21$ |
| LSNO [30] | 0.994 | $1.020 \pm 0.387$ | $0.248 \pm 0.095$ | $7.72 \pm 3.19$ |
| EGFormer [40] | $\mathbf{0.831}$ | $0.945 \pm 0.386$ | $0.182 \pm 0.084$ | $7.63 \pm 3.26$ |

## 4.3 Shallow water equations on the rotating sphere

The shallow water equations are a system of partial differential equations that describe the dynamics of a thin fluid layer on a rotating sphere. As such, this system is a useful model for geophysical fluid flows such as atmospheric dynamics or ocean dynamics [21, 5]. We use the dataset as presented in [6]. Initial conditions are sampled from a Gaussian random process and reference solutions are generated on the fly using a spectral solver. The solutions are computed at a resolution of $128 \times 256$ on a Gaussian grid, on a sphere which matches the Earth's radius and angular velocity. The target solution is set at a lead time of 30 minutes, which requires 12 time-steps of the numerical solver to compute the target due to time-stepping restrictions. A detailed explanation is provided in Appendix C.2.

All architectures are trained using solutions output by a traditional spectral solver, with 12 classical time-steps steps to provide the single prediction step at a learning rate of $10^{-3}$. All of the above is carried out using the ADAM optimizer [27] and using a ReduceLROnPLeateau learning rate scheduler. On an NVIDIA A6000 GPU, training took approximately between 24 and 34 minutes depending on the model. To evaluate the models, we compared the classically-solved solution with model predictions after a single prediction step and five autoregressive prediction steps.

Table 3: Numerical results on the shallow water equations on the rotating sphere. Validation errors are reported after a single and after five autoregressive steps.

| Model | Training loss ↓ | Validation loss ↓ | $L_1$ error ↓ | $L_2$ error ↓ |
|---|---|---|---|---|
| | | 1 step | | |
| $\mathbb{R}^2$ Transformer | 0.051 | $0.051 \pm 0.002$ | $0.148 \pm 0.003$ | $0.196 \pm 0.005$ |
| $S^2$ Transformer | 0.003 | $0.003 \pm 0.000$ | $0.042 \pm 0.001$ | $0.054 \pm 0.001$ |
| $\mathbb{R}^2$ Transformer (local) | 0.015 | $0.016 \pm 0.001$ | $0.075 \pm 0.002$ | $0.115 \pm 0.004$ |
| $S^2$ Transformer (local) | **0.003** | $\mathbf{0.003 \pm 0.000}$ | $\mathbf{0.040 \pm 0.001}$ | $\mathbf{0.050 \pm 0.001}$ |
| $\mathbb{R}^2$ SegFormer | 0.087 | $0.085 \pm 0.003$ | $0.214 \pm 0.004$ | $0.281 \pm 0.005$ |
| $S^2$ SegFormer | 0.039 | $0.038 \pm 0.001$ | $0.147 \pm 0.003$ | $0.193 \pm 0.004$ |
| $\mathbb{R}^2$ SegFormer (local) | 0.052 | $0.050 \pm 0.002$ | $0.163 \pm 0.003$ | $0.219 \pm 0.005$ |
| $S^2$ SegFormer (local) | 0.034 | $0.033 \pm 0.001$ | $0.138 \pm 0.002$ | $0.181 \pm 0.004$ |
| | | 5 steps | | |
| $\mathbb{R}^2$ Transformer | - | $36.986 \pm 9.390$ | $3.282 \pm 0.367$ | $5.237 \pm 0.605$ |
| $S^2$ Transformer | - | $0.062 \pm 0.006$ | $0.190 \pm 0.009$ | $0.239 \pm 0.011$ |
| $\mathbb{R}^2$ Transformer (local) | - | $5.667 \pm 0.566$ | $0.933 \pm 0.044$ | $1.987 \pm 0.088$ |
| $S^2$ Transformer (local) | - | $\mathbf{0.033 \pm 0.005}$ | $\mathbf{0.137 \pm 0.010}$ | $\mathbf{0.172 \pm 0.012}$ |
| $\mathbb{R}^2$ SegFormer | - | $0.680 \pm 0.025$ | $0.618 \pm 0.012$ | $0.781 \pm 0.015$ |
| $S^2$ SegFormer | - | $0.154 \pm 0.013$ | $0.304 \pm 0.013$ | $0.386 \pm 0.017$ |
| $\mathbb{R}^2$ SegFormer (local) | - | $0.546 \pm 0.024$ | $0.544 \pm 0.013$ | $0.692 \pm 0.016$ |
| $S^2$ SegFormer (local) | - | $0.104 \pm 0.008$ | $0.250 \pm 0.010$ | $0.319 \pm 0.013$ |

Based on the outcomes of this experiment as reported in Table 3, we highlight the significantly reduced validation loss of the spherical architectures compared to their Euclidean counterparts. This is reflected in the single- and five-step results in the table and visualized in Figure 5, which depicts a drastic reduction in distortion errors towards the poles for the $S^2$ neighborhood Transformer over its Euclidean counterparts. We note that training on larger time-steps caused all transformer models (including the $S^2$ models) to be unstable on autoregressive roll-outs, which non-transformer $S^2$-optimized convolutional, neural-operator models reported in [6, 30] do not seem to suffer from.

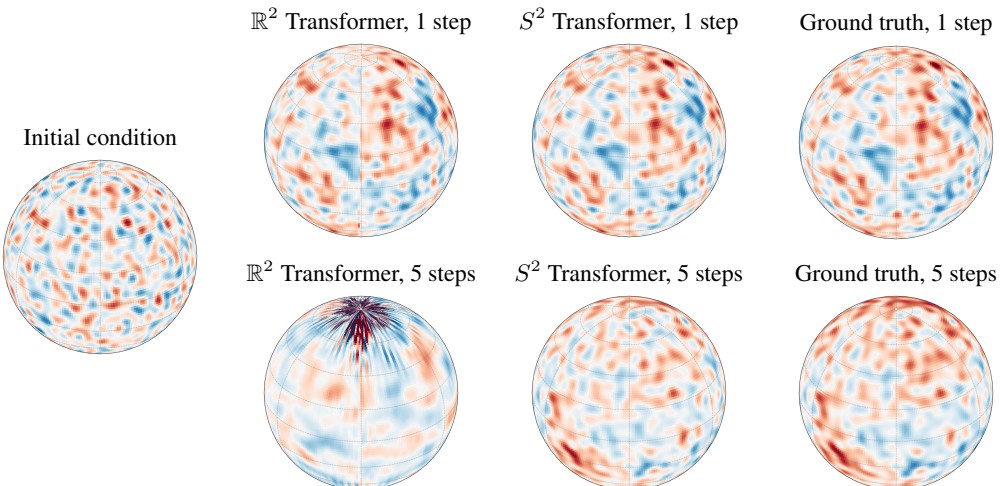

Figure 5: Comparison of single step and five-step autoregressive predictions of both local $\mathbb{R}^2$ and $S^2$ Transformers to a ground truth solution of the shallow water equations.

## 4.4 Ablation study

We conduct an ablation study on the shallow water equations setting to assess the contribution of individual architectural choices. Starting from the general-purpose vision transformer [15], we incrementally introduce spherical convolution, spherical attention, and local attention to evaluate

their individual and combined effects. Training losses and validation metrics after 5 autoregressive steps are reported in Table 4. The results imply that incorporating the inductive biases of spherical geometry and locality into encoding and attention layers is highly effective in obtaining competitive performance on the given task and other geophysical settings. We remark that the changed from learned position embeddings to spectral embeddings decreases the skill as it constitutes a significant reduction in trainable parameters. Moreover, some improvements are only notable as you after multiple autoregressive steps, such as the changes from global $\mathbb{R}^2$ to local $S^2$ attention.

Table 4: Ablation results on the shallow water equations on the rotating sphere. Validation reports are reported for 5 autoregressive steps.

| Model | Training loss ↓ | Validation loss ↓ | $L_1$ error ↓ | $L_2$ error ↓ |
|---|---|---|---|---|
| ViT [15] | 0.463 | $1.202 \pm 0.043$ | $0.805 \pm 0.014$ | $1.013 \pm 0.017$ |
| – patch embeddings
+ $\mathbb{R}^2$ conv. encoder | 0.042 | $6.753 \pm 0.948$ | $1.384 \pm 0.060$ | $2.210 \pm 0.137$ |
| – learned pos. embed.
+ spectral pos. embed. | 0.053 | $40.361 \pm 8.379$ | $3.275 \pm 0.310$ | $5.285 \pm 0.529$ |
| – $\mathbb{R}^2$ conv. encoder
+ $S^2$ conv. encoder | 0.003 | $0.079 \pm 0.011$ | $0.208 \pm 0.013$ | $0.261 \pm 0.016$ |
| – $\mathbb{R}^2$ attention
+ $S^2$ attention | 0.003 | $0.062 \pm 0.007$ | $0.193 \pm 0.011$ | $0.242 \pm 0.013$ |
| – $S^2$ global attention
+ $S^2$ local attention | **0.003** | $\mathbf{0.034 \pm 0.006}$ | $\mathbf{0.141 \pm 0.012}$ | $\mathbf{0.177 \pm 0.014}$ |

# 5 Limitations and future work

The unstructured nature of the neighborhoods in our $S^2$ method requires custom CUDA kernels that are well optimized, and show increased performance over the full attention methods using heavily optimized SDPA kernels available within PyTorch. The NATTEN [23] package used in our baseline comparisons provides optimized neighborhood attention kernels (for a fixed, rectangular neighborhood size), including support for lower precision data types, such as Float16 and BFloat16. Our custom kernel employs vectorization over features and optimized memory access but utilizes 32 bit floating point type. We will investigate if our kernel can benefit from lower precision data types as well. Furthermore, we aim at implementing distributed memory support for both global and neighborhood spherical attention layers, similar to other layers implemented in torch-harmonics.

# 6 Conclusion

We have generalized the attention mechanism in its continuous formulation to the two-dimensional sphere, obtaining an equivariant formulation. Moreover, a spherical variant of neighborhood attention using geodesic distance has been derived. These approaches are implemented as neural operators via quadrature rules for practical spherical discretizations.

In conjunction with other spherical signal-processing techniques, we obtain spherical counterparts of popular Vision Transformer architectures such as the ViT, Neighborhood Transformer, and SegFormer. The effectiveness of our method has been demonstrated for three distinct learning tasks. The new spherical Transformers consistently outperform their Euclidean counterparts, demonstrating the importance of geometrically faithful treatment of such problems. Considering the growing importance of transformer architectures in scientific and engineering domains, we foresee this method becoming a valuable approach for researchers and practitioners seeking to advance the state of the art.

## Acknowledgments

We are especially grateful to our colleague Mauro Bisson for helping us write and optimize CUDA kernels. Moreover, we are grateful to Alexander Keller, J.P. Lewis, and Vishal Mehta for fruitful discussions and proofreading the manuscript.

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

# A   Signals on the sphere

## A.1   Grids and quadrature rules

Our method deals with real-valued signals $u : S^2 \to \mathbb{R}^n$ defined on the unit sphere

$$x(\vartheta, \varphi) = \begin{bmatrix} \sin \vartheta \cos \varphi \\ \sin \vartheta \sin \varphi \\ \cos \vartheta \end{bmatrix}. \tag{8}$$

Processing these signals requires discretizations by means of a sampling scheme. While there are various sampling theorems on the sphere [16, 32], we restrict our discussion to sampling schemes motivated by quadrature rules. Many of the operations discussed in this chapter require evaluating integrals of the form

$$\int_{S^2} u(x) \, \mathrm{d}\mu(x) = \int_0^{2\pi} \int_0^{\pi} u(x(\vartheta, \varphi)) \, \sin \vartheta \, \mathrm{d}\vartheta \, \mathrm{d}\varphi, \tag{9}$$

where $u \in L^1(S^2)$ is an integrable function on the sphere and $\mu(x)$ denotes the Haar measure on the sphere. We approximate this integral using a quadrature rule

$$\int_{S^2} u(x) \, \mathrm{d}\mu(x) \approx \sum_{i=1}^{n_{\mathrm{grid}}} u(x_i) \, q_i, \tag{10}$$

which is characterized by a choice of grid points $\{x_i\}_{i=1}^{n_{\mathrm{grid}}}$ and corresponding quadrature weights $\{q_i\}_{i=1}^{n_{\mathrm{grid}}}$. While our methods are applicable to many types of grids, we typically use the equiangular grid $\{x(\vartheta, \varphi) | \, \forall \vartheta \in \{\vartheta_i\}, \, \varphi \in \{\varphi_j\}\}$ defined by

$$\vartheta_i = \pi i / n_{\mathrm{lat}} \qquad \text{for } i = 0, 1, \ldots, n_{\mathrm{lat}} - 1, \tag{11a}$$

$$\varphi_j = 2\pi j / n_{\mathrm{lon}} \quad \text{for } j = 0, 1, \ldots, n_{\mathrm{lon}} - 1, \tag{11b}$$

where $n_{\mathrm{lat}}$ and $n_{\mathrm{lon}}$ denote the number of grid points in latitude and longitude. The associated quadrature weights are

$$q_{ij} = \frac{2\pi^2}{n_{\mathrm{lat}} n_{\mathrm{lon}}} \sin \vartheta_i, \tag{12}$$

and approximately sums to $4\pi$. This choice of quadrature weights correspond to trapezoidal quadrature weights in spherical coordinates.

## A.2   Loss functions and metrics

For two real-valued signals on the sphere $u, u^* : S^2 \to \mathbb{R}$, we define loss functions for the purpose of regression and classification tasks. All of the losses are integrated over the sphere which is numerically evaluated using a quadrature rule.

$L_p$ **norms**   The $L_1$ distance on the sphere is defined via the $L_1(S^2)$ norm as follows:

$$\mathcal{L}_1[u, u^*] = ||u - u^*||_{L_1(S^2)} = \frac{1}{4\pi} \int_{S^2} |u - u^*| \, \mathrm{d}\mu(x). \tag{13}$$

In the same fashion, the squared $L_2$ distance is defined as

$$\mathcal{L}_2^2[u, u^*] = ||u - u^*||_{L_2(S^2)}^2 = \frac{1}{4\pi} \int_{S^2} (u - u^*)^2 \, \mathrm{d}\mu(x). \tag{14}$$

**Sobolev norm**   The spherical Sobolev $W^{1,1}$ semi-norm is defined as

$$||u - u^*||_{W^{1,1}(S^2)} = \frac{1}{4\pi} \int_{S^2} \left| \frac{1}{\sin \theta} \partial_\phi (u - u^*) \right| + |\partial_\theta (u - u^*)| \, \mathrm{d}\mu(x). \tag{15}$$

The loss $\mathcal{L}_{\mathrm{depth}}$ can be composed as a combination of the Sobolev $W^{1,1}$ semi-norm and the $L_1$ distance:

$$\mathcal{L}_{\mathrm{depth}} = ||u - u^*||_{L_1(S^2)} + \lambda ||u - u^*||_{W^{1,1}(S^2)}, \tag{16}$$

with $\lambda = 0.1$ to approximately match the magnitudes of both components in the depth estimation task.

**Cross entropy loss**  The cross entropy loss $\mathcal{L}_{\text{CE}}$ can be defined as follows

$$\mathcal{L}_{\text{CE}} = -\frac{1}{4\pi} \int_{S^2} \text{CrossEntropy}(u, u^*), \tag{17}$$

for $C$ classes, and the predicted and ground truth functions $u$ and $u^*$. The point-wise cross-entropy is given by

$$\text{CrossEntropy}(u, u^*) = -\sum_{c=1}^{C} u_c^* \log \left( \frac{\exp(\text{logits}(u_c))}{\sum_{i=1}^{C} \exp(\text{logits}(u_i))} \right), \tag{18}$$

and the logits operator is given by

$$\text{logits}(u) = \log \left( \frac{u}{1-u} \right). \tag{19}$$

**Intersection over Union (IoU)**  IoU is a commonly used metric for segmentation quality that captures the correctly guessed segmentation classes normalized by the sum of the correct predictions and all false predictions. The IoU is given by

$$\text{IoU} = \frac{\sum_{c=1}^{C} T_p^{(c)}}{\sum_{c=1}^{C} T_p^{(c)} + F_p^{(c)} + F_n^{(c)}}, \tag{20}$$

where for a given class $c$, $T_p^{(c)}, T_n^{(c)}, F_p^{(c)}, F_n^{(c)}$ denote area fraction associated with true positives, true negatives, false positives, and false negatives, respectively. [1] Those can be computed from the predictions $u_c$ and ground-truth values $u_c^*$ via

$$T_p^{(c)} = \frac{1}{4\pi} \int_{S^2} u_c \cap u_c^* \, \mathrm{d}\mu(x), \tag{21a}$$

$$F_p^{(c)} = \frac{1}{4\pi} \int_{S^2} u_c \cap u_{\neg c}^* \, \mathrm{d}\mu(x), \tag{21b}$$

$$F_n^{(c)} = \frac{1}{4\pi} \int_{S^2} u_{\neg c} \cap u_c^* \, \mathrm{d}\mu(x), \tag{21c}$$

$$T_n^{(c)} = \frac{1}{4\pi} \int_{S^2} u_{\neg c} \cap u_{\neg c}^* \, \mathrm{d}\mu(x). \tag{21d}$$

$$\tag{21e}$$

For each class $c$, those quantities satisfy

$$T_p^{(c)} + F_p^{(c)} + F_n^{(c)} + T_n^{(c)} = 1. \tag{22}$$

**Accuracy (Acc)**  For segmentation evaluation, the segmentation accuracy

$$\text{Acc} = \frac{1}{C} \sum_{c=1}^{C} T_p^{(c)} + T_n^{(c)} \tag{23}$$

is the fraction of true predictions in the overall domain, averaged over all classes $c$.

### A.3  Spherical harmonics

The spherical harmonic functions $Y_l^m(\vartheta, \varphi)$ define a set of orthogonal polynomials on the sphere. They are defined as

$$Y_l^m(\vartheta, \varphi) = (-1)^m c_l^m P_l^m(\cos\vartheta) e^{im\varphi} = \widehat{P}_l^m(\cos\vartheta) e^{im\varphi}, \tag{24a}$$

$$c_l^m := \sqrt{\frac{2l+1}{4\pi} \frac{(l-m)!}{(l+m)!}}, \tag{24b}$$

---

[1]Note that (20) is sometimes referred to as micro-average, where the class averages are computed over numerator and denominator separately. This definition is more stable than it's macro-averaged counterpart, where the average is computed over the ratio instead. For this work, we use the micro-averaged expression.

where $P_l^m(\cos\vartheta)$ are the associated Legendre polynomials. The normalization factor $c_l^m$ is chosen to normalize the spherical harmonics w.r.t. the $L^2(S^2)$ inner product, s.t.

$$\int_{S^2} Y_l^m(x)\,\overline{Y_{l'}^{m'}(x)}\,\mathrm{d}\mu(x) = \delta_{ll'}\delta_{mm'}. \tag{25}$$

# B  Implementation details

## B.1  Discrete realization

With an appropriate quadrature rule applied to the integrands of the continuous formulation, we get the discrete attention

$$\mathrm{Attn}_{S^2}[q,k,v](x) = \sum_{j=1}^{N_{\mathrm{grid}}} \frac{\exp\!\left(q^T(x)\cdot k(x_j)/\sqrt{d}\right)}{\sum_{l=1}^{N_{\mathrm{grid}}} \exp\!\left(q^T(x)\cdot k(x_l)/\sqrt{d}\right)\omega_l}\,v(x_j)\,\omega_j, \tag{26}$$

followed by the discrete neighborhood attention

$$\mathrm{Attn}_{S^2}[q,k,v](x) = \sum_{j=1}^{N_{\mathrm{grid}}} \frac{\mathbb{1}_{D(x)}(x_j)\,\exp\!\left(q^T(x)\cdot k(x_j)/\sqrt{d}\right)}{\sum_{l=1}^{N_{\mathrm{grid}}} \mathbb{1}_{D(x)}(x_l)\,\exp\!\left(q^T(x)\cdot k(x_l)/\sqrt{d}\right)\omega_l}\,v(x_j)\,\omega_j, \tag{27}$$

## B.2  Global spherical attention implementation

Our global spherical attention mechanism leverages PyTorch's native scaled dot product attention (SDPA), where quadrature weights $\omega_i$ are incorporated via the framework's additive attention mask. SDPA applies additive attention masking, i.e. by adding the mask tensor to the $q\cdot k$ logits prior to computing the softmax. We can translate this into a multiplicative weighting of the exponential functions by encoding $\log(\omega_i)$ directly into the mask-a valid approach for strictly positive weights ($\omega_i > 0$).

## B.3  Spherical neighborhood attention implementation

Our implementation for spherical neighborhood attention follows the ideas used to derived discrete continuous convolutions on the sphere (DISCO) by [34]. In this work, a spherical convolution can be described by a contraction of sparse matrix $\psi$, which encodes the geometry of the problem, with a dense input tensor $x$ of shape $B\times C\times H\times W$:

$$y[b,k,c,h,w] = \sum_{h'=0}^{\mathrm{nlat}-1}\sum_{w'=0}^{\mathrm{nlon}-1} \psi[k,h,h',w']\,x[b,c,h',w'+w] \tag{28}$$

Here, $0\le k < K$ and $K$ is the dimension of a chosen set of basis functions which support as well as coefficient values are encoded in $\psi$. The final result of the convolution is obtained by contracting the channels $c$ and basis indices $k$ with a learnable weight tensor.

Note that $\psi$ only depends on the input and output grids (via it's sparsity structure and quadrature weights for the input grid $\omega_{h'}$) as well as the number and type of basis functions. However, $\psi$ does not depend on the number of channels $c$ or any learnable parameters. Therefore, $\psi$ can typically be re-used across many different layers in contemporary neural networks. Additionally, $\psi$ does not depend on $w$ because all grids considered in this work are translationally invariant in longitude direction. This again reduces the memory footprint of DISCO convolution layers.

In order to implement spherical neighborhood attention, we modify equation (28). In this specific case, we can choose $K = 1$ and do not need to store any basis coefficient values, as out attention kernel (5) computes the appropriate neighbor weights dynamically. In this case, $\psi$ is basically just an indicator function $\mathbb{1}[h,h',w']$, which is equal to one for all $|x_i(h,0) - x_j(h',w')|_{S^2} \le \theta_{\mathrm{cutoff}}$ and zero otherwise. We can write:

$$y[b, c, h, w] = \sum_{h'=0}^{\text{nlat}-1} \sum_{w'=0}^{\text{nlon}-1} v[b, c, h', w' + w]$$

$$\times \frac{\mathbb{1}[h, h', w'] \, \exp\left(\sum_{l=0}^{d-1} q[b, l, h, w] \cdot k[b, l, h', w' + w]/\sqrt{d}\right) \omega[h']}{\sum_{h''=0}^{\text{nlat}-1} \sum_{w''=0}^{\text{nlon}-1} \mathbb{1}[h, h'', w''] \, \exp\left(\sum_{l=0}^{d-1} q[b, l, h, w] \cdot k[b, l, h'', w'' + w]/\sqrt{d}\right) \omega[h'']} . \quad (29)$$

As noted, the desire for a weighted spherical neighborhood of our attention model (6) comes with the downside that we cannot leverage pre-existing optimized attention implementations, which do not parameterize quadrature weights or the shape of the neighborhood.

### B.4 Spherical neighborhood attention gradients

Beyond the CUDA implementation of forward attention $A$, we have to derive and implement the backward passes of the model. If we let $q_i := q(x_i)$, $k_j := k(x_j)$ and $\alpha_{ij} := \exp(q_i \cdot k_j) \omega_j$, starting with the Jacobian w.r.t. $q_i$, which requires use of the quotient rule, we define the terms

$$f = \sum_j \alpha_{ij} v_j \qquad\qquad f' = \sum_j \alpha_{ij} k_j$$

$$g = \sum_j \alpha_{ij} \qquad\qquad g' = \sum_j \alpha_{ij} k_j,$$

which combine to form the the Jacobian

$$\frac{\partial A}{\partial q_i} dy_i = \left(f'(dy_i \cdot v_j)g - f(dy_i \cdot v_j)g'\right) g^{-2}$$

$$= \left(\sum_j (\alpha_{ij} k_j (dy_i \cdot v_j)) \sum_j \alpha_{ij} - \sum_j (\alpha_{ij}(dy_i \cdot v_j)) \sum_j (\alpha_{ij} k_j)\right)$$

$$\left(\sum_j (\alpha_{ij})\right)^{-2} \quad (30)$$

Next we will show the Jacobian for $k_j$, which uses the softmax derivative identity

$$\sigma(p)_i = \frac{\exp(p_i)}{\sum_j \exp(p_j)}, \qquad \frac{d\sigma(p)_i}{dp_j} = p_i \delta_{ij} - p_i p_j.$$

With this identity, we can derive the Jacobian for $k_j$

$$\frac{\partial A}{\partial k_j} dy_i = \frac{\sum_j q_i \alpha_{ij}}{\sum_j \alpha_{ij}} \left(dy_i \cdot v_j - \frac{\sum_j (\alpha_{ij}(dy_i \cdot v_j))}{\sum_j \alpha_{ij}}\right) \quad (31)$$

The Jacobian for $v$ is simply

$$\frac{\partial A}{\partial v_j} dy_i = \frac{\sum_j \alpha_{ij} dy_i}{\sum_j \alpha_{ij}} \quad (32)$$

### B.5 Spherical neighborhood attention CUDA implementation

With the discrete neighborhood attention formulation, we implement the necessary forward and backward routines in a custom CUDA extension to ensure reasonable performance for our $S^2$ neighborhood experiments. The implementation is thread parallel over the neighborhood points $k_j$, with output points ($q_i$) parallelized over thread blocks. Due to this choice of parallelism, the softmax is computed in two steps, computing the necessary $\max(q \cdot k)$ required to avoid overflow in the exponential first, and finalizing the softmax sum per query point in the second step. This pattern is repeated for the gradient kernel, which additionally fuses $q, k, v$ gradients into a single kernel. This parallelization strategy diverges from traditional optimized attention kernels, and we leave further optimizations and restructuring of this kernel to future work.

## C  Experiments

### C.1  Experimental setup

Standard convolutional networks and Vision Transformers are designed for planar data and struggle with spherical inputs due to distortions introduced by equirectangular projections. These distortions, particularly near the poles, break translation equivariance and lead to inconsistent feature extraction, limiting performance on tasks requiring spherical understanding. To address these limitations, we implement spherical generalizations of these architectures.

**Baselines**  We compare against standard Euclidean models applied to equirectangular projections, including the Vision Transformer (ViT) [15] and SegFormer [38]. These models operate on regular 2D grids and perform attention or convolution uniformly.

ViT employs convolutional patch embedding with kernel and stride size $2 \times 2$ as well as global attention. To allow flexibility with input sizes not divisible by the patch stride, we also evaluate a modified Transformer that uses overlapping $3 \times 3$ patch embeddings and replaces the decoder's reshaping with bilinear interpolation. Both Transformers use 4 attention layers, an embedding dimension of 128, GELU activation, and either layer or instance normalization. The Transformer neighborhood variant uses localized attention with a $7 \times 7$ kernel. ViT uses learnable positional embeddings, while Transformer makes use of spectral positional embeddings. The standard Transformer has 531,968 parameters, while the neighborhood variant has 532,480 parameters.

SegFormer follows a hierarchical architecture with four stages. It uses embedding dimensions $[16, 32, 64, 128]$, attention heads $[1, 2, 4, 8]$, and layer depths $[3, 4, 6, 3]$. Convolutions use $4 \times 4$ kernels, and models include MLP ratios of 4.0, dropout rates of 0.5, and drop path rates of 0.1. Neighborhood versions of SegFormer apply local attention using a $7 \times 7$ attention kernel. The resulting standard SegFormer has 688,321 parameters, and its neighborhood variant has 689,265 parameters.

**Spherical counterparts**  To evaluate the effectiveness of the spherical counterparts, we compare against spherical models that mirror their respective Euclidean baselines, with matching embedding dimensions and parameter counts. The Spherical Transformer follows the ViT structure but replaces patch embeddings and global attention with discrete-continuous convolutions and attention on the sphere. The localized variant restricts the attention window to neighborhoods with a geodesic cutoff radius of $7\pi/(\sqrt{\pi}\,n_{\text{lat}})$ radians, where $n_{lat}$ is the latitudinal dimension, matching the effective receptive field of the planar $7 \times 7$ attention window at the equator. Both the global and localized spherical Transformer models have 531,968 parameters.

Spherical SegFormer models replicate the hierarchical encoder-decoder structure of the Euclidean Seg-Former, with identical stage-wise dimensions, heads, and depths. They employ discrete-continuous convolutions for the encoding-decoding steps. Down-sampling and up-sampling use overlap-based patch merging and spherical convolutions to preserve structure without distortion. The spherical SegFormer has 606,113 parameters, consistent across both standard and neighborhood versions.

All spherical models employ GELU activation, instance normalization, and spectral positional embeddings. For the spherical discrete-continuous convolutions, we set the number of piecewise linear basis functions equivalent to the parameter count of their Euclidean counterparts kernels (e.g., 9 basis functions for a $3 \times 3$ kernel).

### C.2  Datasets

**Segmentation of spherical data**  To facilitate future usage of the Stanford 2D3DS Dataset [1], we report additional details regarding its usage. The semantic labels is provided as a json file in the GitHub repository, while the image data itself is hosted at `https://cvg-data.inf.ethz.ch/2d3ds/no_xyz/`. Each picture's segmentation information `chair_9_conferenceRoom_1_4` is labeled to account for both the type of object (e.g., *chair*, *ceiling*) and the room information (e.g., `conferenceRoom_1_4`), where we drop the additional room information leaving us with only 14 classes. We additionally note that the semantic labels were built such that one label 855309 overflowed to 3341 (due to uint8 handling of integers), which is ignored when using NumPy 1.x,

Table 5: Overview of spherical models and Euclidean counterparts.

| Model | Encoder/Decoder type | Attention type | Parameter count |
|---|---|---|---|
| $\mathbb{R}^2$ Transformer | 2D convolution | Global | 531,968 |
| $\mathbb{R}^2$ Transformer (local) | 2D convolution | $7 \times 7$ window | 532,480 |
| $S^2$ Transformer | $S^2$ convolution | Global | 531,968 |
| $S^2$ Transformer (local) | $S^2$ convolution | Geodesic window | 531,968 |
| $\mathbb{R}^2$ SegFormer | 2D convolution | Hierarchical | 688,321 |
| $\mathbb{R}^2$ SegFormer (local) | 2D convolution | $7 \times 7$ window | 689,265 |
| $S^2$ SegFormer | $S^2$ convolution | Hierarchical | 606,113 |
| $S^2$ SegFormer (local) | $S^2$ convolution | Geodesic window | 606,113 |

but as of NumPy 2.x, this overflow is caught, which we fixed in our implementation, but needed to account for the assumed overflow in the semantic labels json file.

**Depth estimation on spherical data**    We adapt our approach to predict per-pixel depth values from single-view panoramic images. We leverage the Stanford 2D3DS dataset's panoramic depth maps, which are stored as 16-bit PNGs with a maximum measurable distance of $128\,\mathrm{m}$ and a sensitivity of $1/512\,\mathrm{m}$. Each depth value represents the Euclidean distance from the camera center to a point in the scene. To recover the metric depth from the RGB channels of the PNG images, we use the following formula:

$$d = \frac{R + 256G + 256^2 B}{512} \tag{33}$$

where $R$, $G$, and $B$ denote the 8-bit values of the red, green, and blue channels, respectively. This linear transformation preserves the original 16-bit precision while accommodating PNG storage constraints. To address invalid or distorted regions inherent in the equirectangular projection we apply a mask covering 15% of the latitude at both poles to eliminate areas with severe distortion. The remaining invalid pixels, indicated by the value $2^{16} - 1$, are filtered out. Additionally, the data is standardized to have zero-mean, unit-variance distributions across all batches.

**Shallow water equations on the rotating sphere**    The shallow water equations on the rotating 2-sphere model a thin layer of fluid covering a rotating sphere. They are typically derived from the three-dimensional Navier-Stokes equations, assuming incompressibility and integrating over the depth of the fluid layer. They are formulated as a system of hyperbolic partial differential equations

$$\begin{cases} \partial_t \varphi + \nabla \cdot (\varphi u) = 0 & \text{in } S^2 \times (0, \infty) \\ \partial_t (\varphi u) + \nabla \cdot T = S & \text{in } S^2 \times (0, \infty) \\ \varphi = \varphi_0 & \text{on } S^2 \times \{t = 0\}, \\ u = u_0 & \text{on } S^2 \times \{t = 0\}. \end{cases} \tag{34}$$

The state $(\varphi, \varphi u^T)^T$ contains the geopotential layer depth $\varphi$ (mass) and the tangential momentum vector $\varphi u$ (discharge). In curvilinear coordinates, the flux tensor $T$ can be written using the outer product as $\varphi u \otimes u$. The right-hand side contains flux terms such as the Coriolis force. A detailed treatment of the SWE equations can be found in e.g. [21, 5, 33].

Training data for the SWE is generated by randomly generating initial conditions and advancing them in time using a classical numerical solver. The initial geopotential height and velocity fields are realized as Gaussian random fields on the sphere. The initial layer depth has an average of $\varphi_{\text{avg}} = 10^3 \cdot g$ with a standard deviation of $120 \cdot g$. The initial velocity components have a zero mean and a standard deviation of $0.2 * \sqrt{\varphi_{\text{avg}}}$. The parameters of the PDE, such as gravity, radius of the sphere and angular velocity, we choose the parameters of the Earth. Training data is generated on the fly by using a spectral method to numerically solve the PDE on an equiangular grid with a spatial resolution of $256 \times 512$ and timesteps of 150 seconds. Time-stapping is performed using the third-order Adams-Bashford scheme. The numerical method then computes geopotential height, vorticity and divergence as output.

This data is z-score normalized and the modes are trained using epochs containing 256 samples each. To optimize the weights, we use the popular Adam optimizer with a learning rate of $2 \cdot 10^{-3}$.

Table 6: Numerical results for the equivariance test. The test is performed on equiangular grids of various resolutions.

| | global $S^2$ attention | | local $S^2$ attention | |
| Resolution | Interpolation error | Equivariance error | Interpolation error | Equivariance error |
| --- | --- | --- | --- | --- |
| $13 \times 24$ | $0.321 \pm 0.096$ | $1.097 \pm 0.658$ | $0.360 \pm 0.040$ | $0.879 \pm 0.119$ |
| $25 \times 48$ | $0.190 \pm 0.047$ | $0.674 \pm 0.372$ | $0.212 \pm 0.018$ | $0.541 \pm 0.063$ |
| $49 \times 96$ | $0.105 \pm 0.040$ | $0.220 \pm 0.118$ | $0.106 \pm 0.010$ | $0.222 \pm 0.030$ |
| $97 \times 192$ | $0.052 \pm 0.028$ | $0.070 \pm 0.038$ | $0.045 \pm 0.007$ | $0.086 \pm 0.014$ |
| $193 \times 384$ | $0.018 \pm 0.015$ | $0.021 \pm 0.015$ | $0.014 \pm 0.003$ | $0.028 \pm 0.005$ |
| $385 \times 768$ | - | - | $0.003 \pm 0.001$ | $0.008 \pm 0.002$ |

## D    Equivariance test

To verify the approximate equivariance of the spherical attention formulation we study the equivariance error

$$\frac{\left\| \Psi_{R^{-1}} \, \text{Attn}_{S^2}[\Psi_R \, q, \Psi_R \, k, \Psi_R \, v](x) - \text{Attn}_{S^2}[q, k, v](x) \right\|_{L^2(S^2)}}{\left\| \text{Attn}_{S^2}[q, k, v](x) \right\|_{L^2(S^2)}}, \tag{35}$$

where $\Psi_R$ is the representation corresponding to the rotation $R \in SO(3)$, passively rotating a function $f \in L^2(S^2)$, such that $\Psi_R f(x) = f(R^{-1}x)$. Equation (35) measures the approximate equivariance by comparing the result of applying the attention mechanism to a rotated signal and rotating it back to the result of directly applying it to the original signal. In the perfectly equivariant case, this error is exactly $0$.

To perform this test numerically, we sample a Gaussian process on the sphere, which is defined by random, normally distributed spectral harmonic coefficients up to degrees $l = m = 12$. This process is repeated three times to generate random inputs $q(x), k(x)$ and $v(x)$. The resulting functions are in turn rotated numerically by applying the rotation $R$ to the grid on which the functions are represented and subsequently interpolating the result onto the original grid. We use cubic interpolation, while making sure that the periodicity at the boundary is respected. The spherical attention mechanism is applied and the result is rotated back using the same method. The outcome is then compared to the attention mechanism directly applied to the original signals. As this procedure incurs an interpolation error due to the numerical application of the rotation, we compare the results to the interpolation error $\|\Psi_{R^{-1}} \Psi_R \, a(x) - a(x)\|_{L^2(S^2)} / \|a(x)\|_{L^2(S^2)}$, where the same rotation and it's inverse are numerically applied to the output of the attention mechanism $a(x) = \text{Attn}_{S^2}[q, k, v](x)$.

We perform the test for the rotation $R$ defined by the Euler angles $\alpha = \pi/3, \beta = \pi/5. \gamma = \pi/2$, but find these results remain qualitatively the same for other choices of $R$. The results are averaged over 32 different random initializations of $q, k$ and $v$ and both mean and standard deviation are reported in Table 6. We observe that the equivariance error decays roughly at the same rate as the interpolation error incurred when rotating the signals, which indicates that the method is indeed approximately equivariant. Moreover, for the global $S^2$ attention, we find that the equivariance error is only slightly larger than the interpolation error and roughly twice as large as the interpolation error in the case of the local $S^2$ attention. While it is not possible to clearly separate the contribution of the interpolation error to the equivariance error, this indicates that the equivariance error is at worst twice that of the interpolation error.

