# OpenReview forum: "Attention on the Sphere"
_NeurIPS.cc/2025/Conference — NeurIPS 2025 poster_

### Official Review · Reviewer_nUh3 · 2025-07-02

**Clarity:** 2
**Significance:** 3
**Originality:** 3
**Rating:** 3
**Confidence:** 4

**Summary:**

The authors introduce a generalized attention mechanism for spherical domains, enabling Transformers to effectively process data on the 2D sphere. By incorporating numerical quadrature weights into the attention mechanism, an approximately rotationally equivariant and geometrically faithful spherical attention is obtained, offering stronger inductive biases and better performance than Cartesian methods. To improve scalability and locality, the authors additionally propose neighborhood attention that limits interactions to geodesic neighborhoods, reducing computational cost while maintaining symmetry. They provide optimized CUDA implementations and demonstrate their method’s effectiveness across three tasks: shallow water simulation, spherical image segmentation, and depth estimation. The spherical Transformers consistently outperform planar baselines, emphasizing the benefits of geometric priors for learning on spherical domains.

**Questions:**

1. Are the $\mathbb{R}^{2}$ and $S^{2}$ versions of Transformer, Transformer (local), SegFormer, SegFormer (local), and SphereUFormer trained using the same input data? If yes, is this input appropriate and optimal for validating the advantages of the proposed method?
2. Can the authors clarify the input representation used for the Stanford 2D3DS dataset? Specifically, how is the data preprocessed or adapted to fit the spherical model’s input requirements?
3. Given the abstract and introduction, there is an expectation to see validation on diverse domains such as atmospheric physics, cosmology, and robotics. Can the authors provide additional experimental results or validation on datasets from these or other relevant domains?
4. How does the proposed method compare quantitatively and qualitatively to existing spherical CNNs and geometric deep learning models?
5. How generalizable is the model to data with varying resolutions or noise characteristics?

**Ethical Concerns:**

["NO or VERY MINOR ethics concerns only"]

**Final Justification:**

While I appreciate the authors’ effort to generalize local and global attention mechanisms to the spherical domain, I remain unconvinced by their justification for not providing stronger empirical validation.

The authors state that their goal is not to present a new state-of-the-art architecture for spherical data, but rather to offer a generalization of attention mechanisms to the spherical setting. However, I believe that a method aiming to generalize such foundational building blocks should be accompanied by more substantial experimental evidence to demonstrate its effectiveness and practical relevance on more tasks like classification.

Moreover, the authors’ assertion that it is “impossible to say” whether their method performs better than others in certain settings seems evasive. While it may not be possible to make universal claims, it is certainly reasonable to expect a focused empirical analysis that explores strengths and limitations across representative tasks. Without this, the contribution risks being overly theoretical or speculative.

I encourage the authors to strengthen their work by including:
1. Comparative evaluations against relevant baselines (e.g., Spherical CNNs, existing transformer variants for spherical data) on more tasks.
2. A more grounded discussion of the method’s limitations and applicable scenarios.

This would significantly improve the clarity and impact of the contribution.

**Limitations:**

Yes

**Quality:**

2

**Strengths And Weaknesses:**

Strengths:

1. The motivation is clear, and the problem is important.
2. The paper is well written and clearly organized.
3. Validation is done on three different tasks.

Weaknesses:

1. The literature review in the related work section is insufficient. It is mentioned in the introduction that spherical convolutional neural networks and geometric deep learning models have been used for learning on the sphere, the paper does not provide a detailed review or comparison of these prior works. Additionally, the relationship between the proposed method and neighborhood Transformers is not clear.
2. The suitability of the Stanford 2D3DS dataset for the proposed approach is questionable. It is not clear what input representation was used during training with this dataset, raising concerns about whether the data aligns well with the model’s intended application.

---

> ### Author Rebuttal · Authors · 2025-07-31
>
> **W1 - Review of Prior work**
> We agree with the reviewer that our review of prior work was incomplete and could have been more extensive. Since the review, we have added additional references to equivariant graph transformers as suggested by reviewer WwPf ([1,2,3,4]) along side additional baselines (SphericalCNN[5] and EGFormer[6]) to the related work sections and results sections.
>
> Our focus in the manuscript is on extending the popular attention mechanism used by transformer & segformer architectures to the spherical domain. Given this goal, we did not set out to beat the state of the art, but instead ensure that our technique is applicable across several image and simulation applications, with benefits over naive euclidean approaches. Despite this, we agree that baselines are helpful in estimating the performance of the approach and have added SphericalCNN[5] and EGFormer[6] to the segmentation and depth estimation problems.
>
> **W2 - Suitability of the 2D3DS dataset**
> The 2D3DS dataset provides problems which require mapping spherical signals to other spherical signals, in contrast to other datasets which often deal with estimating pose or other features. This makes depth estimation and segmentation a natural choice given that the architecture is aimed at operator learning (i.e. mapping spherical signals to other spherical signals). Moreover, the 2D3DS dataset has been a popular choice for manuy related works such as [7].
>
> For both segmentation and depth estimation, the input is an RGB image that was downsampled to 128x256 and 256x512 to reduce the computational cost and enable both non-local and non-spherical attention mechanisms as baselines, which otherwise become intractable. We agree that this could have been exposed better in the text and have edited the manuscript accordingly.
>
> **Q1 - Are all models trained with the same input/outputs. Is this appropriate to validate the method?**
> Yes - all of the architectures are trained in the same manner to ensure comparability. Moreover the same loss formulation is used for all architectures, and architectures are kept as close to eachother in terms which makes this a the good test to evaluate the advantage of the spherical attention formulation.
>
> One weakness of the dataset is that the segmentation and depth content closer to the poles was less variable. For the shallow water equations, a large source of visually noticeable artifacts arise at the poles, where our equivariant methods have no issues. Despite this we see advantages with our methods for all three datasets, which validates the approach.
>
> Finally, the dataset has been used by many previous works giving a good comparison, at both a visual and quantitative (e.g., IoU) level for the applicability of our method. Metrics have been added to verify the validity of the method beyond just the validation loss.
>
> **Q2 - 2D3DS data preprocessing**
> Yes - The dataset is downsampled from it’s original 4K representation to a more suitable resolution of 128x256 on an equiangular grid on the sphere. The target in the depth estimation case is a logarithm of the depth whereas in the segmentation case it is a class label enumerated by an integer number. The input data is z-score normalized to ensure proper training of the architectures. We agree that this should have been clarified better and have edited the manuscript accordingly
>
> **Q3 - Other application areas**
> The depth estimation and segmentation datasets were chosen due to their relevance in robotics, whereas the Shallow Water Equation example was chosen due to the relevance of the dataset in geophysics and especially atmospheric physics. Regarding cosmology, we are not aware of a suitable image-based dataset, however there is a big body of work that deals with signal processing of spherical data in cosmology. Given the short period of time of the rebuttal phase we chose to focus on adding additional baselines and an equivariance test (see other reviews). We adapted the language in the manuscript to only mention cosmology in the context of possible application domains.
>
> **Q4 - Comparison against spherical CNNs**
> We have implemented the SphericalCNN [5] architecture and the EGFormer [6] and added them as additional baselines to the tables provided below (Table 1 and 2). In our initial, time-limited, attempt to implement the baselines, the segmentation IoU performance is worse, which is visually notable in the segmentation results. Our technique relies on a suitable quadrature rule to enable numerical integration on the given input grid, and to use an icosohedral representation would require numerical integration rules for the different grid as well as updates to our neighborhood implementation to enable gathering of the icosohedron within the geodesic neighborhood. That said, one could imagine plugging into an adapted neighborhood attention mechanism into the SphericalCNN architecture after the input grid is projected to the icoshedral grid.
>
> *Table1*: Updated table for the segmentation problem
> | Model | Training loss ↓ | Validation loss ↓ | IoU ↑ | Acc ↑ |
> |--|--|--|--|--|
> | **Results on 128×256 data** | | | | |
> | S² Transformer   | 0.330 | 1.02 ± 0.572  | 0.578 ± 0.126 | 0.961 ± 0.015 |
> | S² SegFormer  | 0.306 | **0.705 ± 0.307** | 0.662 ± 0.102 | 0.970 ± 0.011 |
> | ℝ² SegFormer (local)  | 0.289  | 0.817 ± 0.339 | 0.624 ± 0.103 | 0.966 ± 0.011 |
> | **S² SegFormer (local)**  | 0.321  | 0.706 ± 0.303  | **0.658 ± 0.100**| **0.970 ± 0.011**|
> | EGFormer   | **0.203**       | 1.29 ± 0.487   | 0.586 ± 0.106  | 0.962 ± 0.012 |
> | **Results on 256×512 data**  | | | | |
> | Spherical CNN | 0.605 | 0.809 ± 0.287 | 0.590 ± 0.099  | 0.963 ± 0.012    |
> | ℝ² SegFormer (local, large) | 0.626 | 0.812 ± 0.891 | 0.692 ± 0.158 | 0.972 ± 0.018 |
> | **S² SegFormer (local, large)** | **0.515** | **0.656 ± 0.719**    | **0.728 ± 0.136**| **0.976 ± 0.015**|
>
> *Table 2 - Depth Estimation*
> We have updated the depth estimation data with new baselines (EGFormer & SphericalCNN). Please find the updated values below:
>
> | Model | Training loss ↓ | Validation loss ↓ | L₁ error ↓  | W¹,¹ error ↓ |
> |---|----|--|---|--|
> | ℝ² Transformer | 0.954 | 0.982 ± 0.386 | 0.215 ± 0.084 | 7.67 ± 3.22 |
> | S² Transformer | 0.978 | 0.993 ± 0.381 | 0.234 ± 0.086 | **7.59 ± 3.23** |
> | ℝ² Transformer (local) | 1.014 | 1.023 ± 0.370  | 0.237 ± 0.079 | 7.86 ± 3.17 |
> | S² Transformer (local) | **0.978** | 0.976 ± 0.354   | 0.209 ± 0.068       | 7.68 ± 3.12 |
> | ℝ² SegFormer | 1.037 | 0.980 ± 0.371 | 0.169 ± 0.058       | 8.11 ± 3.22 |
> | S² SegFormer | 1.005 | **0.935 ± 0.369**  | **0.165 ± 0.057** | 7.70 ± 3.23 |
> | ℝ² SegFormer (local)        | 1.055   | 1.002 ± 0.374          | 0.174 ± 0.061       | 8.27 ± 3.25          |
> | S² SegFormer (local)        | 1.014           | 0.948 ± 0.368          | 0.174 ± 0.061       | 7.75 ± 3.21          |
> | **Additional baselines**    |  | | |  |
> | Spherical CNN   | 1.019| 1.098 ± 0.412          | 0.315 ± 0.123       | 7.83 ± 3.21          |
> | LSNO  | 0.994  | 1.020 ± 0.387 | 0.248 ± 0.095       | 7.72 ± 3.19          |
> | SUNet  | 1.006 | 0.985 ± 0.352 | 0.220 ± 0.061       | 7.65 ± 3.11          |
> | EGFormer | 0.831 | 0.945 ± 0.386 | 0.182 ± 0.084       | 7.63 ± 3.26          |
>
> **Q5 - Varying input resolutions and noise**
> As noted in Table 1 (segmentation results) we tried two resolutions with the local S2 segformer against the local R2 segformer, and both resolutions had similar evaluation scores to their lower resolution counterparts. The continuous formulation of the architecture makes the method applicable to any reasonable resolution and sampling scheme which allows accurate approximation of the integrals in the attention formulation. Moreover, our implementation of sparse attention (geodesic circle) on the sphere is equivalent with standard cartesian blocked neighborhood attention from a memory and computational complexity standpoint and thus scales favorably compared to full attention as resolution grows. Regarding performance with noise: The 2D3DS dataset is composed of RGB images of the real world and shows that our models are robust against noise found in non-synthetic images. That said, explicitly evaluating noise performance, was beyond the scope of our work, although we have no reason to expect it to perform differently than a traditional Transformer or Segformer model on euclidean data.
>
> [1] Fuchs, F., Worrall, D., Fischer, V., & Welling, M. (2020). SE(3)-transformers: 3d roto-translation equivariant attention networks. Advances in neural information processing systems, 33, 1970-1981.
>
> [2] Liao, Yi-Lun, and Tess Smidt. "Equiformer: Equivariant Graph Attention Transformer for 3D Atomistic Graphs." The Eleventh International Conference on Learning Representations.
>
> [3] Xu, Y., Chen, D., Liu, K., Zakharov, S., Ambrus, R. A., Daniilidis, K., & Guizilini, V. C. (2024). SE(3) Equivariant Ray Embeddings for Implicit Multi-View Depth Estimation. The Thirty-Eighth Annual Conference on Neural Information Processing Systems.
>
> [4] Chatzipantazis, E., Pertigkiozoglou, S., Dobriban, E., & Daniilidis, K. (2023). SE(3)-Equivariant Attention Networks for Shape Reconstruction in Function Space. Int. Conference Learning Representations ICLR.
>
> [5] Chao Zhang, Stephan Liwicki, Sen He, William Smith, Roberto Cipolla: HexNet: An Orientation-Aware Deep Learning Framework for Omni-Directional Input. IEEE Trans. Pattern Anal. Mach. Intell. 45(12): 14665-14681 (2023) [2] Y. Lee, J. Jeong, J. Yun, W. Cho, and K.-J. Yoon, “Spherephd: Applying cnns on a spherical polyhedron representation of 360deg images,” in Proceedings of the IEEE/CVF Conference on Computer Vision and Pattern Recognition, 2019, pp. 9181–9189.
>
> [6] Zelin Zhang, Tao Zhang, KediLI, Xu Zheng. EGFormer. arXiv preprint arXiv:2505.14014, 2025
>
> [7] Jeremy Ocampo, Matthew A. Price, Jason D. McEwen. Scalable and Equivariant Spherical CNNs by Discrete-Continuous (DISCO) Convolutions. arXiv preprint arXiv:2209.13603, 2022•arxiv.org

---

> > ### Comment · Reviewer_nUh3 · 2025-08-05
> >
> > The authors have addressed the concerns raised in my previous review through explanations and edits. However, I recommend a more in-depth analysis of the experimental results. Specifically, why does the proposed method outperform existing approaches or its planar counterparts? The current explanation, which relies primarily on the authors’ stated motivations, does not convincingly account for the improved performance.
> >
> > Be sure to paste your answer here.

---

> > > ### Author Response · Authors · 2025-08-05
> > >
> > > If the reviewer is referring to general performance of S2 vs R2 counterparts: This stems from the spherical nature of the data and its proper treatment. This is especially evident from the ablations (Table 4 in the manuscript) where replacing euclidean components with spherical counterparts yields improvements. Best highlighting the reasons for our improvements are the experiments on the spherical shallow water equations, where purely planar approaches introduce significant artifacts at the poles. This is where the geometry effects matter the most and euclidean approaches typically fail.
> > >
> > > If the reviewer had something else in mind, regarding the in-depth analysis, we kindly ask for clarification, so we may address it.

---

> > > > ### Comment · Reviewer_nUh3 · 2025-08-06
> > > >
> > > > By “in-depth analysis,” I meant the following:
> > > >
> > > > 1. Why does the proposed method outperform existing alternatives, such as Spherical CNNs?
> > > > 2. Why is each component necessary?
> > > >
> > > > Only the second question is addressed through the ablation study. I said the provided explanation is not very convincing for me because I’d like to understand the underlying reasons beyond just geometric effects. Additionally, I’d like to understand in which tasks the proposed method would be a particularly effective or preferable choice. Will the proposed method always a better choice is the input data is defined on the sphere?

---

> ### Author Response · Authors · 2025-08-06
>
> We thank the reviewer for the clarification.
>
> Our work is motivated by the popularity of vision transformers in computer vision problems and Scientific ML problems such as weather prediction. Our aim is to present a generalization of the local and global attention mechanisms to the spherical domain, *not* to present a novel SOTA architecture for spherical domains. As such, we believe that the most appropriate baselines are comparable vision transformers, which we took as starting point for our transformers.
>
> Whether the method can be expected to universally work better for certain types of spherical data than e.g. Spherical CNNs - this is impossible to say. We expect the method to work well in situations where vision transformers [1] and especially neighborhood attention transformers [2] are effective for euclidean data. This includes depth estimation, segmentation, classification etc. If a spherical vision transformer is desired, this paper describes how to achieve it and provides efficient implementations of the respective modules.
>
> The components in our architecture are simply spherical counterparts to the operations encountered in the Vision Transformer [1] and Segformer. For instance, patch embeddings can be understood as a special case of a convolution, which is why we chose to replace them with spherical convolutional embeddings. As such, our architectures are kept as close to their euclidean counterparts as possible. Beyond this we are not aware of other ways to motivate or justify the individual components. We would also like to point out that works such as [1] do not really justify components beyond this, either.
>
> [1] Dosovitskiy, Alexey, Lucas Beyer, Alexander Kolesnikov, Dirk Weissenborn, Xiaohua Zhai, Thomas Unterthiner, Mostafa Dehghani et al. "An image is worth 16x16 words: Transformers for image recognition at scale." arXiv preprint arXiv:2010.11929 (2020).
>
> [2] Hassani, Ali, Steven Walton, Jiachen Li, Shen Li, and Humphrey Shi. "Neighborhood attention transformer." In Proceedings of the IEEE/CVF conference on computer vision and pattern recognition, pp. 6185-6194. 2023.

---

### Official Review · Reviewer_WwPf · 2025-07-03

**Clarity:** 3
**Significance:** 3
**Originality:** 3
**Rating:** 5
**Confidence:** 5

**Summary:**

The authors start with a continuous formulation of attention for the entire sphere as well as for neighborhoods defined by the geodesic distance. The continuous kernel integrals are discretized using quadrature and neighborhoods defined as above. The architecture is a modified ViT (but a Segformer can be used as well) having spherical convolutions for patch encoding, spherical multi-head attention, and a decoder with a bilinear spherical interpolation. Position encoding is spectral using the spherical harmonic basis.
Results on the author's spherical version of ViT and Segformer outperform transformers defined on the regular grid of the sphere in segmentation, depth estimation, and the shallow water problem.

**Questions:**

Q1 (and W3): The authors do not mention what they do with the positional encodings. If they add them or concatenate them to the features then the invariance formula in line 117 does not hold anymore because $q$ and $k$ do not behave like scalars (except the $L=0$ order harmonics). The rest have to be multiplied with a $W_q$ that is the product of a radial function times the spherical harmonics basis as described, in the references above and here:
https://alchemybio.substack.com/p/spherical-equivariant-graph-transformer

Q2 (W4): Clarify the notion of continuous.

Address all other weaknesses.

**Ethical Concerns:**

["NO or VERY MINOR ethics concerns only"]

**Final Justification:**

The authors addressed all my concerns. They recognized that they cannot handle higher than scalar types in inputs and outputs in their layers. They convinced me that the community needs such an approach. Their rebuttal and final remarks are exemplary in their honesty and style. I recommend accept.

**Limitations:**

none identified.

**Paper Formatting Concerns:**

none.

**Quality:**

3

**Strengths And Weaknesses:**

Strength: The paper handles in a correct way the discretization of the sphere by using neighborhoods defined by the geodesic thershold. Such a representation does not suffer from the singularity of the equirectangular grid or icosahedral domains.

Weaknesses: The paper has several weaknesses which I will list here and I would be eager to see the authors' responses and how the paper could be improved.

W1: The authors claim as their main contribution the continuous formulation of the transformer. However, their implementation is the discrete sum of neighboorhood $k$ based on the geodesic distance. This is nothing else than a graph transformer with edges only between points where the geodesic distance is less than the $\theta_{cutoff}$.

W2: Related to W1, the authors misrepresent the literature in lines 81-84. All SO(3) equivariant transformers like the ones below have the spherical attention as a subcase, I will explain below.

Fuchs, F., Worrall, D., Fischer, V., & Welling, M. (2020). SE(3)-transformers: 3d roto-translation equivariant attention networks. Advances in neural information processing systems, 33, 1970-1981.

and their implementation in Alphafold2.

Liao, Yi-Lun, and Tess Smidt. "Equiformer: Equivariant Graph Attention Transformer for 3D Atomistic Graphs." The Eleventh International Conference on Learning Representations.

W3: The authors ignore the processing of the positional encoding in the implementation.  The paper addresses only the scalar-valued version of an equivariant graph transformer. There are papers which combine scalar features (like the features in this paper) and spherical encodings like the equivariant perceiver I/O, for example, see 3.3.2 in:

Xu, Y., Chen, D., Liu, K., Zakharov, S., Ambrus, R. A., Daniilidis, K., & Guizilini, V. C. (2024). SE(3) Equivariant Ray Embeddings for Implicit Multi-View Depth Estimation. The Thirty-Eighth Annual Conference on Neural Information Processing Systems.

W4: The notion of continuous as used in the paper it is not in the sense of a neural operator as cited in [25]. Requirement of a neural operator (see Table 1 in [25]) is to to "query the output at any point". That would be achieved by a decoder that has as an input a query spherical coordinate vector and conditioned on the attention output compute the segmentation or depth value at this point on the sphere. This is, for example, done in the equivariant attention decoder in

Chatzipantazis, E., Pertigkiozoglou, S., Dobriban, E., & Daniilidis, K. (2023). SE(3)-Equivariant Attention Networks for Shape Reconstruction in Function Space. Int. Conference Learning Representations ICLR.

The current paper though uses a bilinear interpolator in the decoder keeping a discrete structure.

W5: The paper is limited to scalar valued functions failing to address applications like vector field prediction on the earth or even optical flow on images. In contrast, the SO(3) equivariant transformer papers can handle both vector  valued functions on any graph defined by the geodesic threshold as well as the processing of the spherical embedding.

W7: The paper claims that the icosahedral discretization "do not take into account the rotational symmetry of the sphere, but rather discrete permutations on the resulting graph". If the icosahedral approaches would have used the same positional encoding as the authors here they would also account for rotation symmetry.

W8: While the authors compare with ShapeUformer in the segmentation task, they do not do in the depth prediction although the ShapeUformer paper contains results (with different metrics).

W9: ShapeUFormer contains depth estimation comparisons with several other approaches like Panoformer, EGformer, SFSS, HexRUnet, HealSWIN, Elite360D.

In summary, the paper lacks in novelty and technical soundness. It needs a significant enhancement to be elevated from a subcase of other papers to a paper competing in scope and properties with the state of the art.

---

> ### Author Rebuttal · Authors · 2025-07-31
>
> First of all, we would like to thank the reviewer for taking their time and providing us with this review. In the following we address the reviewers concerns:
>
> **W1 - Lack of novelty, only a graph transformer, no equivariance for vector-valued**
>
> The reviewer argues that this is only a graph transformer where edges are restricted to nodes within a certain geodesic distance from each other. This is not the case, as a graph transformer does not impose a quadrature rule on the summation imposed from a continuous formulation. A consequence of this is that a graph transformer will have permutation equivariance within the nodes inside this cutoff radius. This is not the case here - we break this permutational equivariance and get SO(3) equivariance - a notable difference and a strong inductive bias. We have emphasized this point in the manuscript to make it clearer.
>
> Moreover, the reviewer cites a number of works that construct SE(3) equivariant transformers based on the equivariant SE(3) Graph attention. While the equivariant Graph attention does indeed ensure SE(3) and therefore SO(3) equivariance, a fundamental difference is that the Equiformer and related approaches mainly consider features/point clouds as inputs whereas we consider entire functions that are densely sampled on the sphere. Moreover the argument that this is a special case of aforementioned architectures does not hold. For instance, their depth-wise product (DTP) ensures equivariance by introducing a weight parameterized by the pairwise distance of features. In contrast , we make use of the proper discretization of the invariant Haar measure to obtain an equivariant formulation. This results in a pre-determined integration weight dependent on the absolute location. As such this formulation cannot be obtained as a special casse from the equiformer.
>
> Finally, we would like to emphasize that there is significant value in realizing special cases of architectures and motivating them from a different theoretical perspective. Moreover, we provide efficient implementations and experimental evaluations that this approach is valid. As such this work presents a valuable addition to the literature, regardless of whether it can be derived from another architecture.
>
> **W2 - misrepresented related works, S2 attention as subcase of SE(3)**
>
> We thank the reviewer for bringing these works to our attention. We believe these works are relevant and should have been added to the manuscript. We have added [1,2,3,4] to the related work section of our manuscript.
>
> **W3 - Ignoring positional embedding, scalar-valued equivariant graph transformer**
>
> In contrast to equivariant graph transformers, we consider functions on the sphere and not features. A natural way of processing the positional embeddings in an equivariant manner is to treat them as inputs, therefore rotating them alongside the input if a rotation is applied. We have clarified this in the text. Regarding vector-valued functions see W5.
>
> **W4 - Clarify notion of continuous**
>
> The notion of continuous refers to all blocks being defined in the continuous domain, such that discretizations of the architecture can be derived at any resolution/grid that do not alter the parameterization. As such, the interpolation is not problematic as it is well-defined for any resolution, given that the discretizations are reasonable. There exist other Neural Operator architectures such as the Convolutional Neural Operator [5] which make heavy use of interpolation. Moreover, the decoder does not require this operation - it is added to avoid aliasing as is common in many modern convolutional architectures. We have clarified this in the manuscript.
>
> **W5 - Equivariance of vector-valued inputs**
>
> We do consider signals defined on the surface of the sphere, rather than features which denote positions or directions in euclidean space. As such rotations are applied as passive rotations to the input signals rather than to the outputs of the functions as well (as would be required). Upon popular request, we have added an equivariance test which demonstrates the approximate equivariance (see Table 3). This test considers vector-valued signals on the sphere which are rotated before the attention is applied and then rotated back. The L2 errors remain within a factor 2 of the interpolation incurred by rotating the signals and interpolating them onto the new grid and then reversing this operation. We have added this equivariance test alongside plots to visually demonstrate the equivariance of the attention mechanism (including vector-valued signals).
>
> *Table 3*: Upon suggestion from multiple reviewers , we have added an equivariance test. This test takes a Gaussian random linear combination of spherical Harmonics to create a vector-valued noisy signal on the sphere, which is cut off at l=12, m=12.Equivariance errors are reported alongside interpolation errors that arise from applying the rotations discretely. All errors are reported in terms of the L2 loss on the sphere.
>
> |  | S2 Transformer | | Neighborhood S2 Transformer | |
> |---|----|----|----|-----|
> | Grid Size | Interpolation |   Equivariance   | Interpolation   | Equivariance   |   |
> | 8x16 | 53.913 | 74.934 | 29.588 | 61.374 |
> | 16x32 | 28.630 | 47.374 | 23.750 | 40.656 |
> | 32x64 | 18.268 | 32.687 | 14.621 | 29.052 |
> | 64x128 | 11.470 | 18.350 | 8.852 | 15.690 |
> | 128x256 | 6.354 | 9.617 | 4.763 | 8.678 |
> | 256x512 |  - |  - | 2.397 | 4.123 |
>
> **W6 - Positional embeddings could fix the icosahedral approaches**
>
> The icosahedral approaches that are referenced in the paper do use graph-based attention and rely on its properties to achieve equivariance[6]. However, this is limited to discrete rotations which map nodes of the icosahedron back to nodes of the icosahedron, irrespective of the positional embeddings. This is stated in their work [6]. In contrast, our method only achieves approximate equivariance (see Table 3), however for arbitrary rotations in SO(3).
>
> **W7 - SphereUFormer not applied to depth**
>
> We took the results for SphereUFormer directly from their paper and used only metrics which are comparable. However, we have added two additional baselines to the depth example.
>
> **W9 - Lack of baselines compared to SphereUFormer**
>
> Upon request from the reviewers we have added SphericalCNN [2] alongside EGFormer [3] to the results section (See Table 2 and Table 1 in he rebuttal for reviewer nUh3). We would like to emphasize that the goal of the paper is not to introduce a novel architecture but rather demonstrate how the attention mechanism is correctly adapted to spherical data. This is why in our experiments we use the same transformer architectures formulated on $R^2$ and demonstrate improvements that can be achieved by formulating the attention mechanism correctly. We expect that this “building block” may find its way into other architectures dealing with spherical data.
>
> *Table 2 - Depth Estimation*
> We have updated the depth estimation data with new baselines (EGFormer & SphericalCNN). Please find the updated values below:
>
> | Model | Training loss ↓ | Validation loss ↓ | L₁ error ↓  | W¹,¹ error ↓ |
> |---|----|--|---|--|
> | ℝ² Transformer | 0.954 | 0.982 ± 0.386 | 0.215 ± 0.084 | 7.67 ± 3.22 |
> | S² Transformer | 0.978 | 0.993 ± 0.381 | 0.234 ± 0.086 | **7.59 ± 3.23** |
> | ℝ² Transformer (local) | 1.014 | 1.023 ± 0.370  | 0.237 ± 0.079 | 7.86 ± 3.17 |
> | S² Transformer (local) | **0.978** | 0.976 ± 0.354   | 0.209 ± 0.068       | 7.68 ± 3.12 |
> | ℝ² SegFormer | 1.037 | 0.980 ± 0.371 | 0.169 ± 0.058       | 8.11 ± 3.22 |
> | S² SegFormer | 1.005 | **0.935 ± 0.369**  | **0.165 ± 0.057** | 7.70 ± 3.23 |
> | ℝ² SegFormer (local)        | 1.055   | 1.002 ± 0.374          | 0.174 ± 0.061       | 8.27 ± 3.25          |
> | S² SegFormer (local)        | 1.014           | 0.948 ± 0.368          | 0.174 ± 0.061       | 7.75 ± 3.21          |
> | **Additional baselines**    |  | | |  |
> | Spherical CNN   | 1.019| 1.098 ± 0.412          | 0.315 ± 0.123       | 7.83 ± 3.21          |
> | LSNO  | 0.994  | 1.020 ± 0.387 | 0.248 ± 0.095       | 7.72 ± 3.19          |
> | SUNet  | 1.006 | 0.985 ± 0.352 | 0.220 ± 0.061       | 7.65 ± 3.11          |
> | EGFormer | 0.831 | 0.945 ± 0.386 | 0.182 ± 0.084       | 7.63 ± 3.26          |
>
>
> **References**
>
> [1] Fuchs, F., Worrall, D., Fischer, V., & Welling, M. (2020). SE(3)-transformers: 3d roto-translation equivariant attention networks. Advances in neural information processing systems, 33, 1970-1981.
>
> [2] Liao, Yi-Lun, and Tess Smidt. "Equiformer: Equivariant Graph Attention Transformer for 3D Atomistic Graphs." The Eleventh International Conference on Learning Representations.
>
> [3] Xu, Y., Chen, D., Liu, K., Zakharov, S., Ambrus, R. A., Daniilidis, K., & Guizilini, V. C. (2024). SE(3) Equivariant Ray Embeddings for Implicit Multi-View Depth Estimation. The Thirty-Eighth Annual Conference on Neural Information Processing Systems.
>
> [4] Chatzipantazis, E., Pertigkiozoglou, S., Dobriban, E., & Daniilidis, K. (2023). SE(3)-Equivariant Attention Networks for Shape Reconstruction in Function Space. Int. Conference Learning Representations ICLR.
>
> [5] Bogdan Raonić, Roberto Molinaro, Tim De Ryck, Tobias Rohner, Francesca Bartolucci, Rima Alaifari, Siddhartha Mishra, Emmanuel de Bézenac. Convolutional Neural Operators for robust and accurate learning of PDEs. Advances in Neural Information Processing Systems, 2023
>
> [6] Sungmin Cho, Raehyuk Jung, Junseok Kwon. Spherical Transformer . arXiv preprint arXiv:2202.04942, 2022•arxiv.org
>
> [7] Taco S. Cohen, Mario Geiger, Jonas Koehler, Max Welling. Spherical CNNs. arXiv preprint arXiv:1801.10130, 2018
>
> [8] Zelin Zhang, Tao Zhang, KediLI, Xu Zheng.

---

> > ### Comment · Reviewer_WwPf · 2025-08-04
> > **Many points clarified**
> >
> > I sincerely appreciate the honest effort of the authors in responding to all reviews.
> >
> > W1: I agree with the authors that the intent is still to use a regular sampling of the sphere which deserves a special handling and it would not make sense to treat it as a special case of an SO(3) transformer. I do not have any critique about novelty anymore.
> >
> > W3 and W6: I agree with the authors, that similar to Segformer one can build a transformer without positional encoding.
> >
> > W4: I still disagree that the authors’ formulation can be characterized as continuous. The result of a continuous layer is not an arbitrary resolution or subsequent interpolation but the computation of a function at any value of the input. This can be achieved with neural operators, coordinate-based or more commonly named implicit layers, or using a function basis like splines.
> >
> > W5: There is a misunderstanding here about the notion of vector-valued or higher representation types. The authors did supply the network with a vector-valued function but all subsequent layers were done using scalar inputs and outputs, because if the authors had output higher types they would need to employ the right order of Wigner matrices in the attention kernels. Moreover, the nonlinear ReLU should look different for higher order features. The reason that we use higher representation types than scalars (see the dissertations of Taco Cohen and Maurice Weiler) is not because we deal with vector-valued inputs but because the networks become more expressive. We appreciate the experiment in Table 3 but this is only an empirical evaluation of what happens rather than a modification of the network.
> >
> > W9: I think the citation to EGFormer is [8]. I appreciate the effort of the authors.
> > I am not sure whether such an important experimental comparison as in Table 2 can be accepted at the rebuttal stage since spherical networks is an established field.

---

> > > ### Author Response · Authors · 2025-08-05
> > >
> > > We thank the reviewer for the response and the clarification on the original questions. In the following, we would like to address W4 and W5:
> > >
> > > W4 - We would like to point the reviewer to [5], which shows how band-limited function spaces admit a continuous-discrete equivalence to allow for robust operator learning. In this setting, ideal (sinc-) interpolation is used to up-sample signals without changing the frequency content. On the sphere, the upsampling can also be achieved with ideal interpolation via the SHT (i.e. bandlimited interpolation as in [5]). In our work, we chose bilinear interpolation for its lower memory footprint and higher throughput as we do not observe any notable difference in skill between the two methods. We have reverted this change. Moreover, we would like to point out that the upsampling can also be achieved using the discrete-continuous convolutions. As such, all operations in the architecture are neural operators in the sense that they can be evaluated at any point in the domain and also be formulated as continuous operations.
> > >
> > > W5 - Upon rereading the rebuttal and the answer from the reviewer we realize that there is indeed a misunderstanding regarding vector-valued functions and how their equivariance is defined. We do not consider rotations where both the input coordinates (i.e. the grid/function) and the outputs are rotated jointly. Such equivariance could be achieved by the techniques mentioned by the reviewer that are also described in [4] (constraints on the attention weights, retaining direction after activations etc.). We have adapted the manuscript and remark in the paper that we focus on equivariance in the sense that the functions (i.e. input coordinates) are rotated. A generalization to vector valued functions and rotating output features can be achieved by employing techniques as described in [4]. Table 3 and the respective equivariance test refer to equivariance as we understand it in the rest of the manuscript, where only the position is rotated, not the outputs.
> > >
> > > W8 - Indeed, the intended citation was EGFormer. It seems the citation was lost while editing the rebuttal.

---

> > > > ### Author Response · Authors · 2025-08-07
> > > >
> > > > As the discussion phase is coming to an end we would like to kindly ask if this addresses the reviewer's concerns?

---

### Official Review · Reviewer_qKvE · 2025-07-03

**Clarity:** 3
**Significance:** 3
**Originality:** 3
**Rating:** 5
**Confidence:** 3

**Summary:**

The paper proposes an extension of the attention mechanism for spherical domains which is geometry aware and approximately rotationally equi-variant, enabling transformer architectures to process data natively defined on the 2D sphere. The authors derive continuous formulations for both global and neighborhood attention mechanisms on spherical domains, then discretize them using quadrature rules to ensure approximate SO(3) rotational equivariance. The method is implemented as neural operators that can handle arbitrary discretizations of the sphere.

The authors evaluate their approach on three diverse tasks: shallow water equation simulation, spherical image segmentation, and spherical depth estimation, demonstrating consistent improvements over euclidean counterparts. Popular architectures like ViT and SegFormer are systematically adapted for spherical domains by replacing standard attention mechanisms with either global spherical attention or neighborhood spherical attention. Additionally, the encoding and decoding laters are modified to handle spherical data by substituting conventional CNN operations with spherical convolutions, etc.

**Questions:**

- In Table 1, why do S2 Transformers perform worse than R2 transformers for both, local and neighbourhood variants?
- For comparison with SphereUFormer, why double the embedding dimension? - to equate for number of parameters?
- How does the quadrature-based approximation quality degrade with grid resolution? Can you provide empirical evidence of the approximate SO(3) equivariance?

**Ethical Concerns:**

["NO or VERY MINOR ethics concerns only"]

**Final Justification:**

I am satisfied with the authors' response and would recommend Accept.

**Limitations:**

yes

**Quality:**

3

**Strengths And Weaknesses:**

**Strengths**
- The paper provides a rigorous maths derivation by extending attention mechanisms to spherical domains through proper geometric treatment. The continuous formulation followed by quadrature-based discretization is mathematically sound.
- The method achieves approximate SO(3) rotational equivariance, a crucial inductive bias for spherical data that standard Transformers lack. This addresses key limitations when applying planar methods to spherical domains.
- Custom CUDA kernel implementations for both variants, full and local spherical attention will be made available, making the approach practically applicable.
- The continuous formulation makes these architectures neural operators, enabling evaluation on arbitrary sphere discretizations while preserving geometric structure.
- The evaluation maintains similar parameter counts and hyperparameters (embedding dimensions, layer depths) between spherical and Euclidean variants, ensuring fair comparisons that isolate the impact of geometric considerations.
- The inclusion of sample outputs across all three tasks provides valuable visual evidence of improvements, particularly highlighting reduced distortions near poles and better preservation of spherical structure.
- In Table 3 and Fig 4 - all S2 variants beat their R2 counterparts for shallow water equations on the rotating sphere

**Weaknesses**
- The paper lacks thorough computational complexity analysis and scalability comparisons, especially for the custom CUDA implementations versus optimized baselines.
- In some experiments (e.g., segmentation), the improvements over Euclidean baselines are modest. The IoU improvements are often within error margins, raising questions about practical significance. The paper claims that “Across all models, we see the spherical variants outperforming their Euclidean counterparts” - this is not exactly true as S2 Transformers are worse in most metrics in Table 1.
- The evaluation primarily compares against naive euclidean applications to spherical data. Beyond brief mentions of SphereUFormer, comprehensive comparisons with other geometric deep learning approaches would strengthen the claims.
- The experiments are conducted on relatively small datasets (e.g., 1,621 images for segmentation with only 35-40 test samples), limiting the ability to assess generalisation and robustness. Larger-scale evaluations would provide more convincing evidence of the method's effectiveness.

---

> ### Author Rebuttal · Authors · 2025-07-31
>
> First of all we would like to thank the reviewer for taking their time to review our manuscript and providing us with feedback.
>
> **W1 - Complexity & performance analysis**
> At time of submission, the CUDA implementation was still too immature for serious complexity and performance experiments, particularly against optimized baselines like NATTEN or PyTorch’s full attention implementation. Since the submission, we have reworked the parallelization strategy to enable inline softmax and much improved coalesced memory access patterns such that the complexity benefit compared to full attention can be realized. Experiments done for this review show that updated local vs full S2 segformer is about 1.5x faster in our tests. Our implementation is made publicly available and we aim to provide timing information in the revised version of the manuscript.
>
> **W2 - Discuss segmentation and depth similarity across image models**
> We have softened the language for the performance of the S2 models on the segmentation and depth datasets, which are within the experimental bounds of the R2 models. Moreover, we have discovered a bug in the implementation of the standard S2 transformer, which has made the results more consistent. Given the out-performance of the S2 models on the shallow water equations, we assume this similar performance is a result of the lack of detail at the poles of the spherical images, which doesn’t allow the S2 models to outperform the R2 models on depth and segmentation particularly at the poles.
>
> **W3 - More comparisons against other spherical techniques**
> While the focus of the paper is to prove the effectiveness of the spherical attention formulation, we agree that more baselines could be added. Upon suggestion we have added an implementation of Spherical CNNs and EGFormer to our segmentation and depth estimation datasets.
>
> *Table1*: Updated table for the segmentation problem
> | Model | Training loss ↓ | Validation loss ↓ | IoU ↑ | Acc ↑ |
> |--|--|--|--|--|
> | **Results on 128×256 data** | | | | |
> | ℝ² Transformer | 0.325  | 0.985 ± 0.390 | 0.588 ± 0.105 | 0.962 ± 0.012 |
> | S² Transformer   | 0.330 | 1.02 ± 0.572  | 0.578 ± 0.126 | 0.961 ± 0.015 |
> | S² SegFormer  | 0.306 | **0.705 ± 0.307** | 0.662 ± 0.102 | 0.970 ± 0.011 |
> | ℝ² SegFormer (local)  | 0.289  | 0.817 ± 0.339 | 0.624 ± 0.103 | 0.966 ± 0.011 |
> | **S² SegFormer (local)**  | 0.321  | 0.706 ± 0.303  | **0.658 ± 0.100**| **0.970 ± 0.011**|
> | EGFormer   | **0.203**       | 1.29 ± 0.487   | 0.586 ± 0.106  | 0.962 ± 0.012 |
> | **Results on 256×512 data**  | | | | |
> | Spherical CNN | 0.605 | 0.809 ± 0.287 | 0.590 ± 0.099  | 0.963 ± 0.012    |
> | ℝ² SegFormer (local, large) | 0.626 | 0.812 ± 0.891 | 0.692 ± 0.158 | 0.972 ± 0.018 |
> | **S² SegFormer (local, large)** | **0.515** | **0.656 ± 0.719**    | **0.728 ± 0.136**| **0.976 ± 0.015**|
>
> *Table 2 - Depth Estimation*
> We have updated the depth estimation data with new baselines (EGFormer & SphericalCNN). Please find the updated values below:
>
> | Model | Training loss ↓ | Validation loss ↓ | L₁ error ↓  | W¹,¹ error ↓ |
> |---|----|--|---|--|
> | ℝ² Transformer | 0.954 | 0.982 ± 0.386 | 0.215 ± 0.084 | 7.67 ± 3.22 |
> | S² Transformer | 0.978 | 0.993 ± 0.381 | 0.234 ± 0.086 | **7.59 ± 3.23** |
> | ℝ² Transformer (local) | 1.014 | 1.023 ± 0.370  | 0.237 ± 0.079 | 7.86 ± 3.17 |
> | S² Transformer (local) | **0.978** | 0.976 ± 0.354   | 0.209 ± 0.068       | 7.68 ± 3.12 |
> | ℝ² SegFormer | 1.037 | 0.980 ± 0.371 | 0.169 ± 0.058       | 8.11 ± 3.22 |
> | S² SegFormer | 1.005 | **0.935 ± 0.369**  | **0.165 ± 0.057** | 7.70 ± 3.23 |
> | ℝ² SegFormer (local)        | 1.055   | 1.002 ± 0.374          | 0.174 ± 0.061       | 8.27 ± 3.25          |
> | S² SegFormer (local)        | 1.014           | 0.948 ± 0.368          | 0.174 ± 0.061       | 7.75 ± 3.21          |
> | **Additional baselines**    |  | | |  |
> | Spherical CNN   | 1.019| 1.098 ± 0.412          | 0.315 ± 0.123       | 7.83 ± 3.21          |
> | LSNO  | 0.994  | 1.020 ± 0.387 | 0.248 ± 0.095       | 7.72 ± 3.19          |
> | SUNet  | 1.006 | 0.985 ± 0.352 | 0.220 ± 0.061       | 7.65 ± 3.11          |
> | EGFormer | 0.831 | 0.945 ± 0.386 | 0.182 ± 0.084       | 7.63 ± 3.26          |
>
>
> **W4 - 2D3DS dataset size**
> The lack of images is definitely a weakness of the 2D3DS dataset. However, we find that numerous works in the literature use this dataset to evaluate the effectiveness of spherical architectures. We chose to keep our architectures small as the goal of the paper was to evaluate the effectiveness of the spherical attention formulation over the euclidean counterparts, and not to develop a SOTA architecture. We believe that for this limited setting, this is sufficient.
>
> **Q1 - why do S2 transformers perform worse than R2 transformers**
> Since the submission of the manuscript, we have discovered a bug in the implementation of the S2 transformer, which has been fixed. Updated results are shown in Table 1. Both S2 and R2 perform very similarly, especially considering the standard deviation.
>
>
> **Q2 - Why double the embedding dimension in the SphereUformer comparison**
> The results for SphereUFormer were taken directly from their paper and they were carried out on double the resolution. To bring our architecture into a similar parameter count, we doubled the embedding dimension.
>
>
> **Q3 - How does quadrature scale with mesh resolution, and empirical test of SO(3) approximate equivariance**
>
> We thank the reviewer for this excellent suggestion. On equiangular grids such as for 2D3DS, we use the Clenshaw-Curtis quadrature rule, whereas on the Gauss grids (Shallow Water equations), we use Gauss quadrature. Theory suggests that both of these methods have an error that bounded by $\mathcal{O}((2 N )^{-k} / k)$ for a k-times differentiable integrand. We have added a test to estimate the approximate equivariance in Table 3. In this test, equivariance is tested by rotating smooth signals (degree up to l=12, m=12) discretely and applying the attention mechanism to the rotated signal before rotating it back. The result is compared to a direct application of the attention mechanism. We find that the error is dominated by the interpolation error which is incurred when the signal is rotated discretely and observe that the total error is within 1.5x of the interpolation error incurred by the rotation. We have added a more detailed explanation of this test alongside the results to the manuscript.
>
> |                  |            S2 Transformer        |          |      Neighborhood S2 Transformer |    |
> |-------------|-------------------|-------------------|-------------------|-------------------|
> | Grid Size   |   Interpolation   |   Equivariance   |   Interpolation   |   Equivariance   |               |
> |-------------|-------------------|------------------|-------------------|------------------|
> | 8x16        |            53.913 |           74.934 |            29.588 |           61.374 |
> | 16x32       |            28.630 |           47.374 |            23.750 |           40.656 |
> | 32x64       |            18.268 |           32.687 |            14.621 |           29.052 |
> | 64x128      |            11.470 |           18.350 |             8.852 |           15.690 |
> | 128x256     |             6.354 |            9.617 |             4.763 |            8.678 |
> | 256x512     |                 - |                - |             2.397 |            4.123 |

---

### Official Review · Reviewer_dPKj · 2025-07-16

**Clarity:** 4
**Significance:** 4
**Originality:** 4
**Rating:** 5
**Confidence:** 4

**Summary:**

This paper revisits attention but on spherical data. Attention mechanisms are derived clearly and implemented successfully on spherical data. The method outperforms planar counter parts in clear evaluation of the method.

**Questions:**

Minor additional ablation study could improve the paper.
Future work may investigate alternative ways to compute convolutions of the transformer.

**Ethical Concerns:**

["NO or VERY MINOR ethics concerns only"]

**Final Justification:**

I am satisfied with the rebuttal.

**Limitations:**

yes

**Quality:**

3

**Strengths And Weaknesses:**

+ The paper has clear theoretical contribution to attention with spheres
+ An implementation is made available and outperforms planar transformers in evaluation
+ Computational runtime is evaluated

- while the evaluation is fairly complete, the paper would benefit from ablation study which tests equivariant transformations of the sphere
- other spherical convolution method could enhance the experiments. For example 2D3DS is frequently reported in spherical convolution works.
- I am wondering if an icosahedron mesh could help with efficiency issues. See

[1] Chao Zhang, Stephan Liwicki, Sen He, William Smith, Roberto Cipolla:
HexNet: An Orientation-Aware Deep Learning Framework for Omni-Directional Input. IEEE Trans. Pattern Anal. Mach. Intell. 45(12): 14665-14681 (2023)
[2]  Y. Lee, J. Jeong, J. Yun, W. Cho, and K.-J. Yoon, “Spherephd: Applying cnns on a spherical polyhedron representation of 360deg
images,” in Proceedings of the IEEE/CVF Conference on Computer Vision and Pattern Recognition, 2019, pp. 9181–9189.

---

> ### Author Rebuttal · Authors · 2025-07-31
>
> First of all, we would like to thank the reviewer for taking their time and providing us with this review. In the following we address the reviewers concerns:
>
> **W1 - equivariant experiments**
>
> As suggested by the reviewer, we have added an equivariance test to the manuscript. The convergence study is reported in Table 1, which shows equivariance and interpolation convergence for both the full and neighborhood transformer as resolution is increased. We see that the error is dominated by the interpolation error incurred when rotating the signal numerically.
>
> *Table 1 - Equivariance test*
> Upon suggestion from multiple reviewers , we have added an equivariance test. This test takes a Gaussian random linear combination of spherical Harmonics to create a vector-valued noisy signal on the sphere, which is cut off at l=12, m=12.Equivariance errors are reported alongside interpolation errors that arise from applying the rotations discretely. All errors are reported in terms of the L2 loss on the sphere.
> |                  |            S2 Transformer        |          |      Neighborhood S2 Transformer |    |
> |-------------|-------------------|-------------------|-------------------|-------------------|
> | Grid Size   |   Interpolation   |   Equivariance   |   Interpolation   |   Equivariance   |               |
> | 8x16        |            53.913 |           74.934 |            29.588 |           61.374 |
> | 16x32       |            28.630 |           47.374 |            23.750 |           40.656 |
> | 32x64       |            18.268 |           32.687 |            14.621 |           29.052 |
> | 64x128      |            11.470 |           18.350 |             8.852 |           15.690 |
> | 128x256     |             6.354 |            9.617 |             4.763 |            8.678 |
> | 256x512     |                 - |                - |             2.397 |            4.123 |
>
>
> **W2 - spherical convolution & other baselines**
>
> We were able to run the suggested SphericalCNN[1] (icosohedral projection) model on our depth and segmentation datasets, with the results presented in the revised Tables 2 & 3. We additionally implemented the EGFormer model for the segmentation dataset with results also presented in Tables 2 & 3. That said, with the limited time for review, it is likely that both models could be improved. For example, we noted that our EGFormer[2] model experienced significant overfitting when training.
>
> *Table 2 - Segmentation*
> For segmentation, we added two new baselines (EGFormer & Spherical CNN) and reran several S2 and R2 attention-based models to ensure comparability across the models.
> | Model                                    | Training loss ↓ | Validation loss ↓    | IoU ↑           | Acc ↑           |
> |-------------------------------------------|-----------------|----------------------|------------------|------------------|
> | **Results on 128×256 data**               |                 |                      |                  |                  |
> | S² Transformer                            | 0.32939         | 1.02 ± 0.572         | 0.578 ± 0.126    | 0.961 ± 0.015    |
> | S² SegFormer                              | 0.306           | **0.705 ± 0.307**    | 0.662 ± 0.102    | 0.970 ± 0.011    |
> | ℝ² SegFormer (local)                      | 0.289           | 0.817 ± 0.339        | 0.624 ± 0.103    | 0.966 ± 0.011    |
> | **S² SegFormer (local)**                  | 0.321           | 0.706 ± 0.303        | **0.658 ± 0.100**| **0.970 ± 0.011**|
> | EGFormer                                  | **0.203**       | 1.29 ± 0.487         | 0.586 ± 0.106    | 0.962 ± 0.012    |
> | **Results on 256×512 data**               |                 |                      |                  |                  |
> | Spherical CNN                             | 0.605           | 0.809 ± 0.287        | 0.590 ± 0.099    | 0.963 ± 0.012    |
> | ℝ² SegFormer (local, large)               | 0.626           | 0.812 ± 0.891        | 0.692 ± 0.158    | 0.972 ± 0.018    |
> | **S² SegFormer (local, large)**           | **0.515**       | **0.656 ± 0.719**    | **0.728 ± 0.136**| **0.976 ± 0.015**|
>
> *Table 3 - Depth Estimation*
> We have updated the depth estimation data with new baselines (EGFormer & SphericalCNN). Please find the updated values below:
>
> | Model                        | Training loss ↓ | Validation loss ↓     | L₁ error ↓          | W¹,¹ error ↓         |
> |-----------------------------|-----------------|------------------------|---------------------|----------------------|
> | ℝ² Transformer              | 0.954           | 0.982 ± 0.386          | 0.215 ± 0.084       | 7.67 ± 3.22          |
> | S² Transformer              | 0.978           | 0.993 ± 0.381          | 0.234 ± 0.086       |**7.59 ± 3.23**      |
> | ℝ² Transformer (local)      | 1.014           | 1.023 ± 0.370          | 0.237 ± 0.079       | 7.86 ± 3.17          |
> | S² Transformer (local)      | **0.978**       | 0.976 ± 0.354          | 0.209 ± 0.068       | 7.68 ± 3.12          |
> | ℝ² SegFormer                | 1.037           | 0.980 ± 0.371          | 0.169 ± 0.058       | 8.11 ± 3.22          |
> | S² SegFormer                | 1.005           | **0.935 ± 0.369**      | **0.165 ± 0.057**   | 7.70 ± 3.23          |
> | ℝ² SegFormer (local)        | 1.055           | 1.002 ± 0.374          | 0.174 ± 0.061       | 8.27 ± 3.25          |
> | S² SegFormer (local)        | 1.014           | 0.948 ± 0.368          | 0.174 ± 0.061       | 7.75 ± 3.21          |
> | **Additional baselines**    |                 |                        |                     |                      |
> | Spherical CNN               | 1.019           | 1.098 ± 0.412          | 0.315 ± 0.123       | 7.83 ± 3.21          |
> | LSNO                        | 0.994           | 1.020 ± 0.387          | 0.248 ± 0.095       | 7.72 ± 3.19          |
> | SUNet                       | 1.006           | 0.985 ± 0.352          | 0.220 ± 0.061       | 7.65 ± 3.11          |
> | EGFormer                    | 0.831           | 0.945 ± 0.386          | 0.182 ± 0.084       | 7.63 ± 3.26          |
>
>
> **W3 - icosohedral mesh**
>
> The choice of an equiangular discretization has several advantages, including allowing direct comparison against standard R2 attention methods, ease of visualization, and direct use of the data, which is frequently given in an equiangular format. Since submission, we have worked on our CUDA implementation with an improved parallelization strategy that enabled online softmax and coalesced channel memory accesses, which have dramatically improved the forward and backward kernel performance. That said, nothing rules out the use of an icosahedron mesh except for finding an appropriate quadrature rule to enable the discretized spatial integration, and adapting the implementation to use indirect addressing to gather the mesh elements within the geodesic neighborhood. The lower density of points at the poles of the sphere could offset the performance losses due to indirect addressing lookups.
>
> **Q - Alternate ways of convolutions of the transformer**
>
> If the reviewer is referring to the choice of the encoding / decoding approaches in the spherical transformer, they were chosen to ensure equivariance throughout the architecture. In particular, because spherical convolutional patch embeddings can be seen as sparse convolution on the sphere.
>
> **References**
>
> [1] Chao Zhang, Stephan Liwicki, Sen He, William Smith, Roberto Cipolla: HexNet: An Orientation-Aware Deep Learning Framework for Omni-Directional Input. IEEE Trans. Pattern Anal. Mach. Intell. 45(12): 14665-14681 (2023) [2] Y. Lee, J. Jeong, J. Yun, W. Cho, and K.-J. Yoon, “Spherephd: Applying cnns on a spherical polyhedron representation of 360deg
> images,” in Proceedings of the IEEE/CVF Conference on Computer Vision and Pattern Recognition, 2019, pp. 9181–9189.
>
> [2] Zelin Zhang, Tao Zhang, KediLI, Xu Zheng. EGFormer. arXiv preprint arXiv:2505.14014, 2025

---

### Note · Authors · 2025-08-11

We thank the reviewers for their detailed feedback and engagement during the discussion phase.

Through the rebuttal and subsequent clarifications, we have addressed concerns regarding novelty, relationship with existing SO(3)/SE(3) equivariant architectures, treatment of positional encodings, the continuous operator property, and extensions to vector-valued functions. We have added both theoretical clarifications and new experimental baselines (e.g., Spherical CNNs and EGFormer), and presented detailed equivariance tests to support our claims.

Our main contribution is a principled extension of attention and neighborhood attention to spherical domains, enabling robust, approximately rotationally equivariant transformer architectures with demonstrated improvements over their Euclidean counterparts—especially for scientific tasks where geometric fidelity is critical. We provide efficient open-source GPU implementations and have carefully documented all implementation, training, and evaluation procedures for reproducibility.

We acknowledge that broader dataset analyses and further extensions (handling higher representation types, integration into SOTA architectures) remain important directions for future work. Our aim is to provide a modular building block enabling application of transformer architectures to spherical data, and to motivate further research at the intersection of geometric deep learning and scientific ML.

Thank you for considering our work and the extensive clarifications provided during the review process.

---

### Decision · Program_Chairs · 2025-09-17

**Decision:**

Accept (poster)

**Comment:**

The reviewer discussion around Attention on the Sphere was divided, with scores ranging from reject to clear accept. Reviewers highlighted that the paper makes a principled and technically sound extension of attention and neighborhood attention mechanisms to spherical domains, with rigorous derivations, a continuous formulation discretized via quadrature, and efficient CUDA implementations. Empirical results across three diverse tasks consistently show benefits of spherical over Euclidean attention, particularly in the shallow water equation benchmark where geometric fidelity is crucial. Multiple reviewers praised the clarity of the paper, the motivation, and the provision of open-source resources, and were ultimately convinced by the authors’ thorough rebuttal and clarifications on novelty, positional encodings, and additional baselines (including Spherical CNNs and EGFormer).

The main disagreements arose around scope and completeness. One reviewer (initially skeptical) argued the work lacked novelty relative to SE(3)-equivariant transformers and questioned whether the “continuous” claim was justified, though after discussion they recommended acceptance. Another reviewer remained concerned about the limited breadth of experimental validation and the lack of deeper comparative analysis against established spherical architectures, leaving a borderline score. Overall, however, the rebuttal addressed most key concerns, clarified misunderstandings, and added substantive evidence, leading the majority of reviewers to converge on acceptance.